

# Connecting smoke plumes to sources using Hazard Mapping System (HMS) smoke and fire location data over North America

Steven J. Brey[1], Mark Ruminski[2], Samuel A. Atwood[1], Emily V. Fischer[1]

[1]Atmospheric Science, Colorado State University, Fort Collins, 80523, U.S.
[2]NOAA/NESDIS Satellite Analysis Branch, College Park, 20740, U.S.

*Correspondence to*: Steven J. Brey (sjbrey@rams.colostate.edu)

**Abstract.** Fires represent an air quality challenge because they are large, dynamic and transient sources of particulate matter and ozone precursors. Transported smoke can deteriorate air quality over large regions. Fire severity and frequency are likely to increase in the future, exacerbating an existing problem. Using the National Environmental Satellite, Data and Information

Service (NESDIS) Hazard Mapping System (HMS) smoke data for North America for the period 2007 to 2014, we examine a subset of fires that are confirmed to have produced sufficient smoke to warrant the initiation of a U.S. National Weather Service smoke forecast. We find that gridded HMS analyzed fires are well correlated (r = 0.84) with emissions from the Global Fire Emissions Inventory Database 4s (GFED4s). We define a new metric, smoke hours, by linking observed smoke plumes to active fires using ensembles of forward trajectories. This work shows that the Southwest, Northwest, and

Northwest Territories trigger the most air quality forecasts, and produce more smoke than any other North American region by measure of the number of HYSPIT points analyzed, the duration of those HYSPLIT points, and the total number of smoke hours produced. The average number of days with smoke plumes overhead is largest over the north-central U.S. Only Alaska, the Northwest, the Southwest, and Southeast U.S. regions produce the majority of smoke plumes observed over their own borders. This work moves a new dataset from a daily operational setting to a research context, and it demonstrates how

changes to the frequency or intensity of fires in the western U.S. could impact other regions.

## 1 Introduction

North American fires represent a major source of atmospheric pollutants (Wiedinmyer et al., 2006), and they contribute to elevated ground level ozone ($O_3$) and fine particulate matter ($PM_{2.5}$) in the U.S. (Baker et al., 2016; Brey and Fischer, 2016; Jaffe et al., 2008; Park et al., 2007; Saide et al., 2015). Exposure to wildfire smoke has been shown to have negative impacts

on respiratory and cardiovascular health (Dennekamp and Carey, 2010; Haikerwal et al., 2015; Rappold et al., 2011). The relative importance of the contribution of smoke, particularly wildfire smoke, as both an $O_3$ precursor and as a direct source of $PM_{2.5}$ will grow as anthropogenic emissions decline (Russell et al., 2012; Val Martin et al., 2015). In addition to air quality considerations, smoke is also thought to make a major contribution to absorbing aerosols observed throughout the troposphere over North America (Forrister et al., 2015; Liu et al., 2015; Liu, 2014; Washenfelder et al., 2015). Thus

understanding the sources and prevalence of smoke is relevant for air quality, visibility, and radiative forcing calculations.



Millions of hectares of forest burn in North America each year (Randerson et al., 2012), the area burned by wildfires in the western U.S. has increased (Westerling, 2016; Westerling et al., 2006), and models predict this trend will continue in a warmer climate (Hurteau et al., 2014; Keywood et al., 2013; Randerson et al., 2012; Scholze et al., 2006; Yue et al., 2013). Agricultural fires are also a significant source of smoke in North America. Over 1.2 million hectares of cropland are burned in the contiguous U.S. each year (McCarty et al., 2009). Though often smaller in size and shorter in duration, these fires have been shown to increase the abundance of gas and aerosol species that can deteriorate air quality (Akagi et al., 2011; Dennis et al., 2002; Jimenez et al., 2006). There are a growing number of case studies documenting instances of smoke traveling thousands of kilometers to affect atmospheric composition far downwind from the fire locations (Baker et al., 2016; Creamean et al., 2016; Miller et al., 2011; Pfister et al., 2008; Stein et al., 2009; Val Martin et al., 2013a). Case studies demonstrate that regional and long-range transport of smoke causes elevated column and surface concentrations of aerosols and trace gases over extensive (continental) regions of the U.S., and the smoke can have implications for atmospheric composition over relatively long temporal scales (weeks) (Creamean et al., 2016; Park et al., 2003, 2007; Val Martin et al., 2013a; Vedal and Dutton, 2006).

In this paper, we show how smoke data from the National Environmental Satellite, Data and Information Service (NESDIS) Hazard Mapping System (HMS) can be used to provide context for the growing number of case studies linking compromised air quality or visibility to smoke. As an introductory example, Fig. 1 shows that during summer (June – September) over the Contiguous U.S. North Dakota, South Dakota, and Minnesota have more smoke plumes overhead than any other states, averaging between 8 and 12 days per month between June and September. The location with the most smoke overhead is approximately co-located with Fargo North Dakota. Figure 1 also incorporates data from $PM_{2.5}$ monitors. Fargo North Dakota lacks continuous ground level $PM_{2.5}$ observations. Nearby monitors in the north east, south east and west of Minnesota show that approximately 10% of these plume days are days where ground level $PM_{2.5}$ is one standard deviation above average summertime concentrations, approximately one day per summer month in these areas. In contrast, Washington and Oregon, both states with major summertime wildfire activity, average six plume days per month. On ~30% of days with smoke plumes in the atmospheric column (~two days per month), ground level $PM_{2.5}$ concentrations are one standard deviation above the summertime average.

The aggregate view provided by Fig. 1 is important because it cannot currently be produced using chemical transport models. A number of advances have recently been made in estimating the emissions inputs by improving burned area products (*e.g* (Randerson et al., 2012) and combining these with emission factors for a wider range of trace species (*e.g.* (Akagi et al., 2011; Wiedinmyer et al., 2011)). However, incorporating the full suite of emitted species into models and simulating the rapid chemical evolution of smoke remains a challenge (Alvarado et al., 2015), as is proper treatment of injection height (Paugam et al., 2016) and the timing of emissions (Saide et al., 2015). Models are also subject to uncertainty associated with meteorological inputs (Garcia-Menendez et al., 2013). Finally, running and analyzing a chemical transport model at the fine grid resolution appropriate to simulate all the individual smoke plumes of interest from North American fires over the scale of a decade is currently too computationally expensive to be practical. Thus other lenses are needed to



examine how the smoke from North American fires is transported and dispersed in the atmosphere over seasonal and interannual timescales.

The two primary goals of this study are to 1) present the distribution and seasonality of fires that trigger National Weather Service (NWS) smoke forecasts, and 2) develop a regional climatology of smoke transport in the U.S. using operational data from the NESDIS HMS combined with forward trajectory calculations. Based on the subset of fires triggering smoke forecasts, HMS observations of smoke in the atmospheric column and trajectory calculations, we present an estimate of the relative frequency that smoke observed over U.S. regions is associated with fires throughout North America.

## 2 Data

### 2.1 Description of operational fire and smoke products

The HMS is an interactive environmental satellite image display and graphical interface system that was developed by NESDIS. Trained satellite analysts use the HMS to generate a daily operational list of fire locations and outline areas of smoke. As a part of this process, analysts manually compare automated fire detections to the mid-wave infrared (MWIR) images used to produce them to ensure each fire exists (Ruminski et al., 2006). Detections deemed to be false are removed, and fires that are not automatically detected are added manually. Visible satellite imagery is also used by analysts to identify fires that may be too small to be automatically detected, either because they do not produce sufficient heat or because they are obscured by a tree canopy (Rolph et al., 2009). In these cases a smoke plume may be the only indication of a fire. The number of fire detections added manually can be significant. For example, over 50% of the total fire detections were added manually during a 12-month period in 2002-2003 examined by (Ruminski et al., 2006). Land-cover data and power-plant locations compliment satellite imagery to help HMS analysts confirm whether automatic detections are fires (Ruminski et al., 2006). The HMS office makes a distinction between all detected fires (hereafter HMS hotspots) and fires an HMS analyst has confirmed to produce a substantial amount of smoke (hereafter HYSPLIT points).

HYSPLIT points are a subset of the HMS hotspots; they are fire detections where an analyst also visually confirms the presence of smoke using visible satellite imagery. HYSPLIT points are human-vetted because they are used to initialize the NWS smoke forecasts (Rolph et al., 2009; Ruminski et al., 2006). Each HYSPLIT point is assigned a latitude, longitude, date, time, and duration. The locations of HYSPLIT points are estimated to be accurate to within 2-3 km. The start time of smoke emissions is estimated to be accurate to within 1 hour. The accuracy of the duration is a bit more uncertain since many of the fires continue to generate smoke after sunset when visible imagery is no longer available. However, it is believed that the duration accuracy for most HYSPLIT points is within 2 hours. Single fires that produce notable amounts of smoke are associated with a cluster of co-located, or nearly co-located, HYSPLIT points in proportion to the amount of smoke observed. The intended operational consequence of designating HYSPLIT points in proportion to the amount of




smoke observed is to allow the NWS smoke forecast model to generate more smoke for large fires and less smoke for smaller fires.

The HMS office does not make any distinctions between sources of smoke. HYSPLIT points can be associated with agricultural burning, prescribed burning, or wildfires (Ruminski et al., 2006). There is a relationship between the number of
HYSPLIT points and the amount of smoke produced by fires, and the analyst determines this relationship. Large wildfires are represented by dozens of HYSPLIT points spread over many square kilometers. These are typically in the western U.S., Canada, and Alaska. The start times of HYSPLIT points can vary within the cluster of points. Operationally, this serves to represent the variability in the amount of smoke observed at different times of the day. For example, during a large wildfire event analysts may create several HYSPLIT points with a 24-hr duration starting at 08 UTC (middle of the night local time).
They also create another set of HYSPLIT points in the vicinity that are assigned a shorter duration and a start time of 20 UTC (early afternoon local time). The operational intention of this strategy is to force the NWS smoke forecast model to produce more smoke in the afternoon and evening and less overnight to replicate observed diurnal trends. HYSPLIT points can also be proxies for unobservable smoke producing fires. When analysts see a large number of HMS hotspots, but due to cloud cover do not directly observe smoke, they create HYSPLIT points in order to initiate the NWS smoke forecast model.
This occurs most frequently in Kansas, Oklahoma, the Northern Plains (Dakotas), and the lower Mississippi Valley (eastern Arkansas, eastern Louisiana, western Mississippi). The Servicio Meteorológico Nacional (SMN, the Mexican National Weather Service) provides most HYSPLIT points over Mexico. These locations are merged with the HMS product. Occasionally the HMS office will perform fire-detection analysis in parts of Northern Mexico. The archives of smoke and fire locations spans from August 2005 to present day; but the analysis presented here spans 2007–2014. These archives
include text files for HYSPLIT points and HMS hotspots and GIS shapefiles for smoke plumes.

This analysis uses data from 2007-2014 for two reasons. First, prior to April 2006 HYSPLIT points did not have duration or start-time estimates. Second, the HMS implemented a system that automatically generates HYSPLIT points in Northern Canada, Mexico, and Central America in the fall of 2014. This change in procedure resulted in more HYSPLIT points than each of the prior years, and there were many more instances of HYSPLIT points assigned durations of 24 hours.
The majority of fires in Northern Canada are wildfires in boreal forests, so each of these automated HYSPLIT points is assigned a duration of 24 hours whereas prior to this it would have received a mix of 24 hour and lesser durations. The intention of the automated system was to reduce the workload of analysts. A similar automated system was implemented for Alaska in 2009. However, this implementation did not lead to a significant change in the proportion of HYSPLIT points with durations of 24 hours analyzed in North America. Prior to Fall 2014 the HYSPLIT points in Mexico and Central America
were intended to be generated by SMN, but at some point in the years prior to 2014 SMN began performing HYSPLIT point analysis inconsistently. As a result NESDIS developed an automated system based on HMS hotspots. The durations for these HYSPLIT points are estimated using the difference between the latest and earliest time for hotspots aggregated into a 20 km grid.



HYSPLIT points are sometimes analyzed at hours when no visible satellite imagery is available to confirm smoke production. This only occurs when the thermal signal in the MWIR imagery is significant in terms of intensity and duration (*e.g.* strong downslope winds at night can cause significant nighttime fire activity). Wildfires can burn for weeks or even months. For the long-lived wildfire scenario, HMS analysts will get two looks daily at a fire from a given polar satellite. One

look overnight (in the Western U.S. between 08 – 10 UTC) and another in the afternoon (20 – 22 UTC). Analysts will add HYSPLIT points at both of those times. For the nighttime pass, fires are often not burning as actively (fewer hotspots). Analysts will add HYSPLIT points based on the fewer number of hotspots and typically assign them a 24-hr duration (*i.e.* total hours smoke production observed), creating a baseline for emissions. In the afternoon, fires are more active and generate more smoke so analysts will add additional HYSPLIT points based on the afternoon satellite data. In this case, they

only assign durations of 10 or 12 hours. The operational significance of these procedures is to attempt to account for the diurnal variations in smoke production in initializing smoke forecasts.

HYSPLIT Points are analyzed in all 50 U.S. States as well as all Canadian Provinces and Territories. Figure 2 shows the location of all HYSPLIT points analyzed between 2007-2014 shaded by the analyzed duration. Dense clusters of 24-hour duration HYSPLIT points are analyzed in regions with active wildfire seasons (*e.g.* Rocky Mountains, Cascade

Mountains, Northern California, Northwest Territories). The Midwest, Great Plains, and Southeast U.S. have many smaller short-lived fires where most HYSPLIT point durations are 12 hours or less. The smoke observed from small short lived fires in the Mississippi River valley is consistent with other studies that consider burn area from small fires *(Randerson et al., 2012)*.

### 2.2 Description of HMS smoke analysis

After identifying HYSPLIT points, HMS analysts use imagery from multiple NOAA and NASA satellites to identify the geographic extent of smoke-plumes (Rolph et al., 2009; Ruminski et al., 2006). Smoke detection is done with visible-band imagery occasionally assisted by infrared to distinguish between clouds and smoke when possible (Ruminski et al., 2006). Geostationary GOES imagery, with its frequent refresh rate (typically every 15 minutes for each spacecraft), is used almost exclusively for smoke detection, although on rare occasions polar orbiting satellite imagery is used. Given the limitations of

the satellite data (mostly obscuration of smoke due to cloud cover), the number and extent of smoke plumes within this dataset represents a conservative estimate. Smoke is sometimes transported to areas with anthropogenic haze pollution. In some cases the smoke will mix with and become indistinguishable from the anthropogenic haze pollution. The greater the distance traveled by a smoke plume, the more challenging it is to distinguish between smoke and anthropogenic haze. This challenge is particularly pronounced for aged smoke impacting the south-eastern U.S. Due to the limitations of visible

satellite imagery and time constraints, no information about the vertical location or extent of smoke plumes is provided. In 2006 HMS analysts began providing estimates of smoke-plume concentrations (*e.g.* 5 μg/m$^3$, 16 μg/m$^3$, and 27 μg/m$^3$). These plumes of varying concentrations are often nested (*e.g.* 27 μg/m$^3$ within 16 μg/m$^3$ and 16 μg/m$^3$ within 5 μg/m$^3$). Smoke plumes vary in size considerably; small plumes cover areas < 100 km$^2$ and others cover several western states.





Between 5 August 2005 and 21 December 2015 there are only 80 days (~2%) where there are either no smoke plume GIS files available or no smoke-plumes analyzed. Most of these days occur during winter months. For this work, we use the archived GIS smoke-plume files available at the following URL ftp://satepsanone.nesdis.noaa.gov/FIRE/HMS/GIS/.

**2.3 Description of land cover characteristics database**

We assign each HYSPLIT point a land cover type using land-cover classifications from the U.S. Geological Survey (USGS) 2002 North American Land-Cover Characteristics 1 km grid-spacing dataset, created by the National Center for Earth Resources Observation and Science (EROS) as part of the Global Land Cover Characterization Project (Brown, 2016). Land-cover characteristics are assessed using 1 km Advanced Very High Resolution Radiometer (AVHRR) data between 1992 and 1993 using the methods described in (Anderson et al., 1976). For this work, we use the GeoTIFF file projected to latitude-

longitude grid as a geospatial raster (Brown, 2016). The latitudinal extent is 18$^o$N - 72$^o$N and 66$^o$W - 172$^o$W. We assign land-cover classifications to HYSPLIT points based on the nearest grid-point center that is not classified as urban or water. When urban or water is the closet grid-point, we substituted the assigned land-cover type with the most common other land-cover classification in the surrounding 0.06 degrees (~5 km) land-cover grid-cells. If all surrounding grid cell land-cover classifications    are    urban    or    water,    no    land-cover    assessment    is    made.    Data    are    available    at

https://www.sciencebase.gov/catalog/item/535fe572e4b078dca33ae61f.

**3 Comparison to Global Fire Emissions Database Version 4s (GFED4s)**

The number of HYSPLIT points assigned to a fire is intended to be proportional to the amount of smoke observed. We gridded HYSPLIT points and compared them to the Global Fire Emissions Inventory Database (GFED; http://www.globalfiredata.org/data.html) version 4s (Giglio et al., 2013; Randerson et al., 2012; van der Werf et al., 2010).

To compare HMS emissions to GFED4s we count the total duration associated with all HYSPLIT points that fall within each GFED4s grid cell for each month between 2007 and 2014. The units of our new gridded emission inventory are thus duration hours per 0.25$^o$ x 0.25$^o$ grid box, hereafter referred to as HMS smoke production duration hours (SPDH). GFED4s produces emissions estimates in kg of carbon. Figure 3 shows a comparison of the cumulative gridded SPDH and monthly GFED4s kg carbon emissions over North America for June to September 2007 to 2014 (32 months). The dark red areas in Fig. 3 show

the locations where SPDH and GFED4s kg carbon emissions have maximum cumulative values for this time period. Figure 3 shows agreement between the two datasets for the locations of their respective maxima in summer, with some notable differences. For example, GFED4s shows the lower Mississippi river valley as an area with significant fire emissions, but this area is not as prominent in the HMS derived dataset. The HMS dataset also show larger extents of local maximums of fire activity compared to GFED4s (*e.g.* North Central Idaho, North Cascades, Gila National Forest). The spatial agreement

for locations with maximum emissions holds when we compare individual summers between 2007 and 2014 (not shown).



For example, both emission inventories agree on the geographical location and extent of the 2013 Quebec fires, an anomalously high fire year for Quebec.

Figure 3c presents the cumulative June though September 2007 to 2014 SPDH (Fig. 3a) divided by GFED4s kg of carbon (Fig. 3b) for each grid cell. Areas represented by the same color can be interpreted as locations where the total HMS

smoke production duration hours assigned to observed fires is consistent with a given quantity of carbon emitted according to GFED4s. As expected based on varying emission factors for different ecosystems, Fig. 3c shows that the relationship between SPDH and GFED4s kg of carbon varies by several orders of magnitude. The largest ratios are observed in North Dakota and southern Manitoba, the southern Mississippi River Valley, Sinaloa Mexico, and over interior Alaska.

When considering the entire North American domain (15 - 80$^{o}$N, 50 - 170$^{o}$W), summed monthly emissions of both

SPDH and GFED4s kg of carbon time series, we observe that the magnitudes of the two inventories are also correlated temporally. Figure 4 shows the normalized domain summed monthly time series for all months between 2007 and 2014. The Pearson correlation coefficient for the two time series is 0.842, confirming that at large scales the two emission inventories are well-correlated. In 2007 and 2008 the normalized HMS duration hours greatly exceed normalized GFED4s emissions. The difference in 2007 is due to large smoke production in the Northwest and Rocky Mountain regions present in the HMS

data. In 2008 extreme smoke production associated with fires in the Southwest are responsible for the discrepancy. Figure 5 shows the Pearson correlation coefficient for the monthly 8-year time series for each grid cell. Locations with large wildfires (Fig. 2) are well correlated between the two datasets (r > 0.90). Locations with small fires like the Southeast, Southern Plains, and Northern Great Plains have lower correlation values (r < 0.20). The high correlation between SPDH and GFED4s kg carbon demonstrate that the duration field of HYSPLIT points is roughly proportional to the amount of smoke produced

by analyzed fires, though that proportionality varies.

## 4 HYSPLIT point regional distributions

Figure 6 designates 10 U.S. regions that closely align, but do not exactly overlap, with EPA regions. Figure 7a presents the total number of HYSPLIT points aggregated for each region. The largest numbers of HYSPLIT points are identified in the Southwest, Northwest, Northwest Territories, and Southeast regions. Figure 7b shows the total SPDH produced by each

region and the percentage of SPDH that are produced by HYSPLIT points analyzed on cropland. For most regions over the course of the entire year, the percent of SPDH owed to cropland is less than 5%. However, the relative importance of cropland fires has a strong seasonality. Notable exceptions are the U.S. Southeast and the Southern Planes, where the SPDH from cropland > 25%. Figure 8 presents the seasonality of North American HYSPLIT points between 2007-2014, and it shows that the majority of HYSPLIT points are identified between June and August. During these months, the dominant land

cover classification for HYSPLIT points is evergreen needle leaf forest, followed by a nearly equal share of scrubland and mixed forest. When viewed in aggregate, the contribution to HYSPLIT points from cropland is largest in the months of April, September, and October.



Figure 9 shows the locations, seasonality, and duration of HYSPLIT points in the 9 contiguous U.S. regions shown in Fig. 8 between 2007 and 2014. The land cover classification for the seasonality and duration histograms is indicated by color using the same color scheme as Fig. 8. HYSPLIT points with durations of 24 hours are usually associated with wildfires. HYSPLIT points with shorter durations represent smaller fires, and often occur on cropland. At the national scale

shown in Fig. 8, grassland and cropland land cover classifications make up a small proportion of the total HYSPLIT points throughout the year. However as discussed above at the regional scale, grassland and cropland HYSPLIT points can represent a significant fraction or even dominate the total number of HYSPLIT points. Figure 9 shows this is the case for the Great Plains, Midwest, and Southern Plains. These regions also have the fewest HYSPLIT points analyzed in the summer months, whereas regions dominated by evergreen forest have a minimum in the winter. The Southwest has more HYSPLIT

points than any other U.S. region, followed by the Northwest, Southeast, and Southern Plains. The Northeast has the fewest HYSPLIT points, which occur mostly on cropland in southern New Jersey. The Mississippi river valley has some of the most densely analyzed HYSPLIT points in the U.S, and these points are located most commonly on forest and cropland. These points are split between the Southeast and Southern Plains regions. The same HYSPLIT point duration and monthly histograms presented in Fig. 9 are available for all regions in the supplemental information.

15       There are more HYSPLIT points analyzed in the Southern Plains (n=41,846) than there are in the Rocky Mountains region (n=35,371). However, this does not indicate that the Southern Plains generate more smoke, because the number of points does not include information on fire duration (discussed in Sect 3.0). The total smoke produced in a region should be proportional to the number of HYSPLIT points multiplied by their respective durations (Fig. 7b). Most of the fires in the Southern Plains have durations less then nine hours while the most-common duration of HYSPLIT points in the Rocky

Mountains region is 24 hours. Alaska (not shown) is an extreme example. Almost all of the HYSPLIT points in Alaska have 24-hour durations. This is by design, since 2009 all HYSPLIT points in Alaska have been assigned 24-hour durations, which is consistent with the types of fires that occur within the state.

## 5 Smoke-transport analysis

### 5.1 Forward trajectories

Forward HYSPLIT trajectories are initialized from each of the HYSPLIT points over the duration of the fires and they represent likely smoke transport pathways. We initialize HYSPLIT trajectories at three different altitudes that span the range of injection heights expected for North American fires based on a prior analysis. Val Martin et al., 2010 present a climatology of smoke-plume heights by land biome for North America. The smoke-plume heights were estimated with the Multi-angle Imaging SpectroRadiameter (MISR) data, and thus could represent an underestimate of the actual injection

height due to the timing of the MISR overpass. A given fire can inject smoke into the atmosphere at many different altitudes. We initialize each HYSPLIT trajectory start hour at three different altitudes spanning the range of injection heights in Val Martin et al., 2010: 500, 1500, and 2500 m amgl. In total, 3,925,932 trajectories are associated with the 517,214 HYSPLIT



points analyzed between 2007 and 2014. For this analysis, trajectories are only considered for the hours that the calculated height above ground level is > 0. Each trajectory was run using the Global Data Assimilation System (GDAS) 1-degree meteorology data and the Eta Data Assimilation System (EDAS) 40km meteorology data. Further specifics on the meteorology data, the HYSPLIT model, and how we ran trajectories for this analysis can be found in Sect S1 of the

Supplemental Information.

## 5.2 Combined analysis of HYSPLIT points and forward trajectories

To build smoke source-receptor relationships for 10 U.S. regions, we define a "smoke hour" as an hourly latitude-longitude HYSPLIT trajectory location (hereafter trajectory point) that overlaps a HMS smoke plume. We use these smoke hours to represent the relative abundance of probable smoke transport pathways. Smoke plume overlap assessments are made using a

two-step process which reduces the number of trajectories included in the analysis initialized at heights that do not represent the smoke injection height of a given fire. Spatial overlap analysis is performed using the R Sp package *over* function (Pebesma et al., 2016). This code and further information on the procedure used can be found in Supplemental Information Sect S3 and S11.

        1) We confirm that at least one of the first 49 trajectory points (2 days + start hour) overlap a smoke plume analyzed

for the same two dates. If none of the first 49 trajectory points overlap a smoke plume, the trajectory is immediately discarded. If any of the first 49 trajectory points overlap a smoke plume, plume-overlap analysis is performed for the entire trajectory (145 trajectory points). We determine the relevance of a trajectory using two days of smoke plumes because smoke plumes only represent smoke perimeters during daylight hours. By including the next-dates smoke plumes in the test, trajectories associated with fires that start in the afternoon are not evaluated more stringently than trajectories associated with

HYSPLIT points for fires in the morning. The results do not change significantly when validating trajectories with only the first-days smoke plumes.

        2) When trajectories meet our first criteria, a point over polygon calculation is done for each matching date trajectory point and smoke plume. There are two weaknesses to this approach: 1) Smoke-plume boundaries are only representative of smoke-plume perimeters during daylight hours, while trajectories exist at both day and night. 2) Some

HMS smoke plumes on individual days are very large; during extreme-smoke events plumes can cover most of the continental U.S., and thus this criteria is not always meaningful. Smoke plumes with different concentration estimates are often nested. For this analysis these smoke plumes are merged, so a trajectory point can only overlap a single plume and contribute one smoke hour. Each trajectory smoke hour is associated with the source region and land cover classification from its initialization point, allowing a source region and land classification analysis of the total number of smoke hours

impacting or emanating from a region. This methodology does not provide information about smoke concentration ($\mu g/m^3$), and we have not placed additional altitude constraints on the trajectories.



### 5.3 Smoke production and frequency by region

Figure 10 shows the total number of smoke hours produced by and present over each region. Years with elevated fire activity in each region can be easily identified using Fig. 10. This figures provides context for isolated case studies of smoke transport associated with extreme periods including the summer 2013 Quebec wildfires (Laffineur et al., 2014), the summer 2012 wildfires in the Rocky Mountain region (Val Martin et al., 2013b), and the 2008 California wildfires (SW Region) (Gyawali et al., 2009). Supplemental Table S3 shows the total smoke hours produced by and over each U.S. region and the differences between using the GDAS and EDAS meteorology datasets for the trajectory calculations.

### 5.4 Smoke transport climatology: Attribution, age and altitude

Figure 11 shows a summary of smoke transport to contiguous U.S. regions between 2007 and 2014 for the months of June, July, August, and September. The first histogram associated with each region shows the distribution of the age (hour in HYSPLIT trajectory) of smoke hours in the atmospheric column above each region separated by the region of origin. The second histogram associated with each region shows the distribution of the height of the smoke in the column above each region (height of the HYSPLIT trajectories). For a given fire, older smoke is more likely to deliver lower $PM_{2.5}$ concentrations than smoke that is only a few hours old. For example, the Great Plains region has more smoke hours (12.3 million hours) than the Northwest (12.1 million hours) and more than the Southwest (10.1 million); however, the average age of smoke hours over the Great Plains is ~one day older than in the Northwest and Southwest.

On average regions that produce the most smoke hours during the summer have more smoke at lower altitudes (*e.g.* Northwest and Rocky Mountains), and the smoke is located at higher altitudes over regions far downwind of summertime fire activity (*e.g.* Northeast). The impact of our choices of 500, 1500, and 2500 m trajectory starting height is apparent in the panels showing trajectory altitude; the majority of the smoke-hours defined over every U.S. region are between these altitudes. There are clear maxima in the number of smoke hours defined at 500, 1500, and 2500 m over source regions that are not receptors of aged smoke hours.

Regions with the largest number of HYSPLIT points tend to contribute the largest proportion of the total smoke hours within their own borders. For example, the smoke impacting the Southwest originates almost exclusively within the Southwest. Smoke present over the Northwest is nearly equally likely to be associated with fires from the Northwest or Southwest. Regions with comparatively little local fire activity have a diverse set of source regions contributing to their total column smoke budgets; examples include the Southern Plains, the Northeast, and the Midwest. The Northwest, Southwest, and Rocky Mountains dominate most other regions total smoke hours; fires in Canada make a major contribution to smoke hours, particularly over the Midwest and the Northeast. The only regions where the three largest contributors of smoke hours are not the Southwest, Northwest, and Rocky Mountains, are the Southeast, Northeast, the Southern Plains and Alaska (not shown). The only regions that contribute more smoke hours to their own total budget then any outside region are the



Southwest, the Northwest, and the Southeast. When the analysis presented in Fig. 11 is generated with trajectories run using EDAS meteorology data, the results are nearly identical and they are available in Sect S7 of the Supplemental Information.

**5.5 Common regional smoke transport pathways**

The Northwest, Southwest, and Rocky mountains produce more smoke hours than any other U.S. regions (Fig. 10). Figure 12 shows the average smoke-transport pathways for fires in these regions for June to September between 2007 and 2014, and it was generated by summing smoke hours. This figure is not a simple trajectory climatology, rather it is a trajectory climatology when smoke-producing fires occur. Smoke produced by fires in California is transported most frequently over Northern California and Eastern Oregon (Fig. 12b). Based on our analysis of the HYSPLIT points these fires occur primarily in evergreen needle leaf forests. Smoke originating from fires in the Northwest is transported most frequently over Eastern Washington, Eastern Oregon, Northern Idaho, and Montana (Fig. 12a). Smoke from fires in Alaska impact every U.S. and Canadian region (not shown). The dominant transport pathway for the smoke crosses Alaska, the Northwest Territories, and Yukon Territory. Smoke traveling from Alaska to Texas has been observed previously (Morris et al., 2006), but this situation is relatively rare when viewed in an aggregate context (Supplemental Information Fig. S3a).

Figure 13 summarizes the climatology of the abundance of smoke hour transport for U.S. regions between June and September 2007 to 2014. The Northwest, Southwest, and Rocky Mountains are responsible for the majority of U.S. summertime smoke hour production. They are also among the smokiest regions by measure of smoke hours overhead and smoke hour age. These three regions also dominate the total smoke hours overhead for all U.S. regions excluding the Southeast and Alaska.

Figure 1 shows that the U.S. Midwest has more smoke plumes overhead than the Northwest and Southwest while Fig. 13 shows that the Northwest and Southwest are the largest U.S. producers and receptors of smoke hours. This occurs because smoke hours are proportional to total smoke while the plumes in Fig. 1 represent the total count of overhead plumes and contains no information about smoke concentration. This contrast shows that there are many dilute smoke plumes over the Midwest since it is downwind of so many smoke producing regions. The Northwest and Southwest by comparison have fewer plumes overhead but are located much closer to fire activity. Plumes over the Northwest and Southwest represent higher smoke concentrations than the plumes observed over Midwest.

**6 Conclusions**

This work moves a new dataset from a daily operational setting to a research context. We define smoke hours, a quantity designed to be proportional to total column smoke, by linking observed smoke plumes to observed fires using HYSPLIT trajectories. This work shows that the Southwest, Northwest, and Northwest Territories trigger the most air quality forecasts, and produce more smoke than any other North American region by measure of the number of HYSPIT points analyzed, the duration of those HYSPLIT points, and the total number of smoke hours produced (Fig. 3, 10-13). This dataset confirms that



there is a substantial amount of smoke-producing fire activity in the Southeast, particularly along the lower Mississippi River Valley. During the summer wildfire season, the largest smoke source regions are located in the west, while receptor regions for smoke are primarily located in the east. The majority of smoke located over western source regions is less than 24 hours old. During summer, our analysis implies that the receptor regions have very little smoke less than 24 hours old with most

smoke in the column older than 48 hours. The Southeast is a unique exception. There is an abundance of fresh smoke (peak near zero hours in Fig. 11) due to the numerous small fires within the region, and this region receives aged smoke (<48 hours of atmospheric aging) from several upwind regions. Though not the focus of the paper, this dataset shows that the Northeast and Mid Atlantic receive more smoke from fires in Canada than regions in the U.S. Midwest.

We present a smoke transport climatology for the summer wildfire season. Based on our metric of smoke hours, the

U.S. regions that produce the most smoke are the Northwest, Southwest, and Rocky Mountains (Fig. 10 and 13). Heavily populated locations in the eastern U.S. (Northeast, Mid Atlantic, Southeast) are routinely impacted by wildland fire smoke from western regions. Thus changes to the frequency or intensity of fires in the U.S. west or Canada is likely to impact the entire U.S. airshed.

**7 Code availability**

All analysis code associated with this project is available online at the following subversion repository: http://salix.atmos.colostate.edu/svn/smokeSource/. The code for the Python module used to execute HYSPLIT Trajectories is available at the following GitHub repository: https://github.com/samatwood/HYSPLITm/. All figures were creating using R and Python. Scripts used to make figures can be identified by a leading "plot" in script names (*e.g.* R/plotRegionSmokeTotals.R). These scripts explicitly reference the data files used to create the plots.

**8 Data availability**

All data for this project are publicly available at the URLs specified in Sect 2. Formatted analysis level data and the code used to format the raw data described in Sect 2, are available in the subversion code repository.

**9 Author contributions**

S. Brey and E. Fischer prepared the manuscript with contributions from all co-authors. E. Fischer was the leader of this

research project. S. Brey performed the majority of analysis, including writing analysis code, creating figures, and executing HYSPLIT trajectories. M. Ruminski is responsible for creating, describing, and properly interpreting Hazard Mapping System HYSPLIT points and smoke plume data. S. Atwood wrote the Python module used to run HYSPLIT Trajectories and served as our expert for best practices in the use of HYSPLIT source-receptor analysis.




## 10 Competing interests

The authors declare that they have no conflict of interest.

## 11 Acknowledgements

Support for this work was provided by the Environmental Protection Agency Award Number 83588401. Bonne Hotmann
made significant science and style contributions to Fig. 1. We gratefully acknowledge the skill and dedication of NESDIS
Hazard Mapping System analysts; their careful operations are responsible for the dataset presented in this paper. The authors
would like to thank Ana Rappold for her work using HMS, which strengthened and guided the early stages of this analysis,
and for a brainstorming phone call in January 2016 which helped spark the authors interest in performing this work. Parts of
this work were included in Steven Brey's masters thesis. His graduate committee, Elizabeth Barnes, Jeffery Pierce, and
Monique Rocca made this work stronger. The authors gratefully acknowledge Matt Bishop for his help in enabling us to
execute millions of HYSPLIT trajectories on our local computer cluster. The authors gratefully acknowledge the NOAA Air
Resources Laboratory (ARL) for the provision of the HYSPLIT transport and dispersion model and READY website used in
this paper. The authors would also like to thank the core development teams of R and Python for making open source
languages that are free and help advance scientific discovery.

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







**Figure 1: North American smoke climatology for June-September 2007-2014. Blue shading indicates the average days per month HMS smoke plumes are analyzed overhead. Markers show non FRM (Federal Reference Method) PM$_{2.5}$ monitors shaded by the percent of plume days where surface 24-hr average PM$_{2.5}$ is greater than the mean ($\mu$) plus one standard deviation ($\sigma$) value for the monitors mean value in June-September. Only monitors with data on 90% of days between June-September 2007-2014 are presented. IMPROVE monitors are marked by squares; all other monitors from the EPA Air Quality System (AQS) are marked by circles. IMPROVE monitors record data every third day. We make the assumption that smoke plumes are equally likely to occur on measurement days as non-measurement days, and thus multiply the number of ground level smoke-impacted plume days at IMPROVE sites by three. Monitors that fall within white areas are influenced by smoke < 1 day per month. The climatology presented here does not relate to smoke plume or ground level PM$_{2.5}$ concentrations, only the presence of smoke in the column and proportion of those days when surface PM$_{2.5}$ is elevated.**





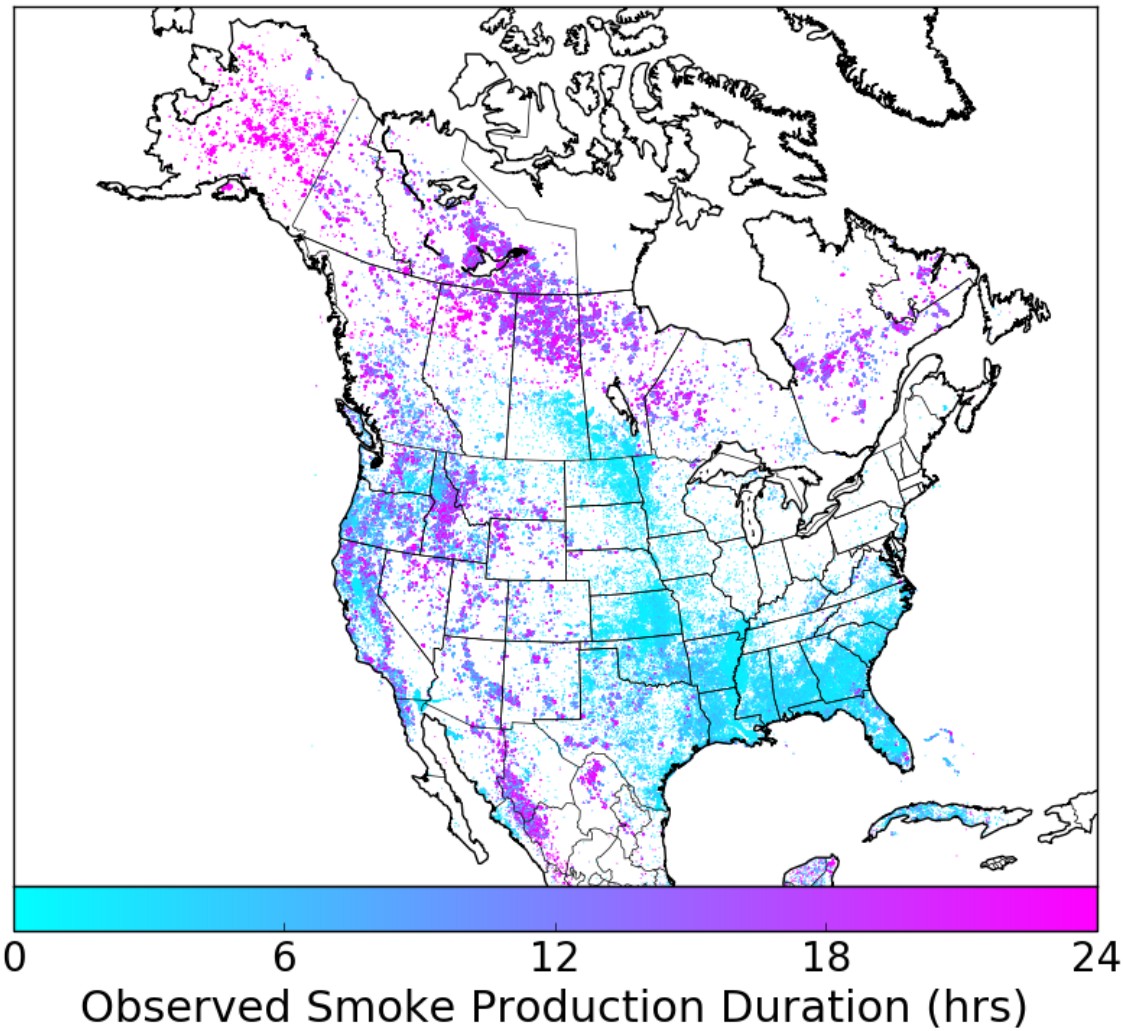

**Figure 2: The locations of all HYSPLIT points analyzed between 2007-2014 shaded by the duration assigned by the analyst. HYSPLIT Points are assigned a duration between 1 and 15, or 24 hours.**



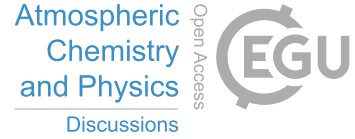

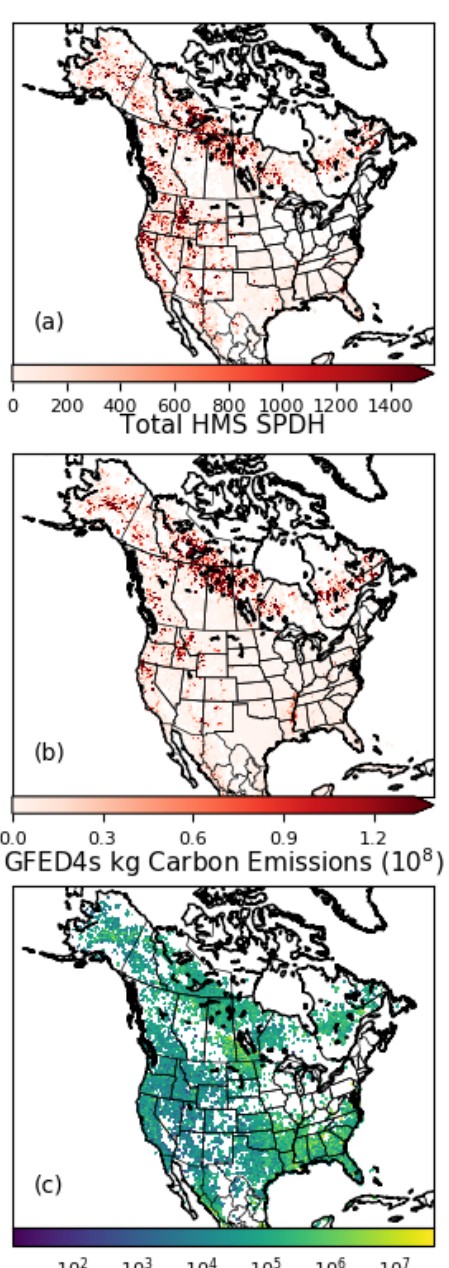




**Figure 3: (a) Cumulative June-September 2007-2014 HMS smoke production duration hours (SPDH) and (b) GFED4s kg carbon per 0.25$^o$ x 0.25$^o$ grid cell. Both emission inventories color bars saturate (darkest red) at the 99th percentile of non-zero values, meaning all values above the 99th percentile value for both datasets are represented by the darkest color on the color bar. (c) Total GFED4s kg of carbon emitted divided by the total HMS smoke production duration hours per grid cell for months June-September 2007-2014. The color bar shows the values on a log$_{10}$ scale where yellow values represent grid cells with more GFED4s kg of carbon per HYSPLIT Point smoke production duration hour and purple indicate fewer. Locations where either GFED4s or HMS emissions are equal to zero are not plotted. The mean kg carbon per HYSPLIT Point smoke production duration hour is 300,123.**

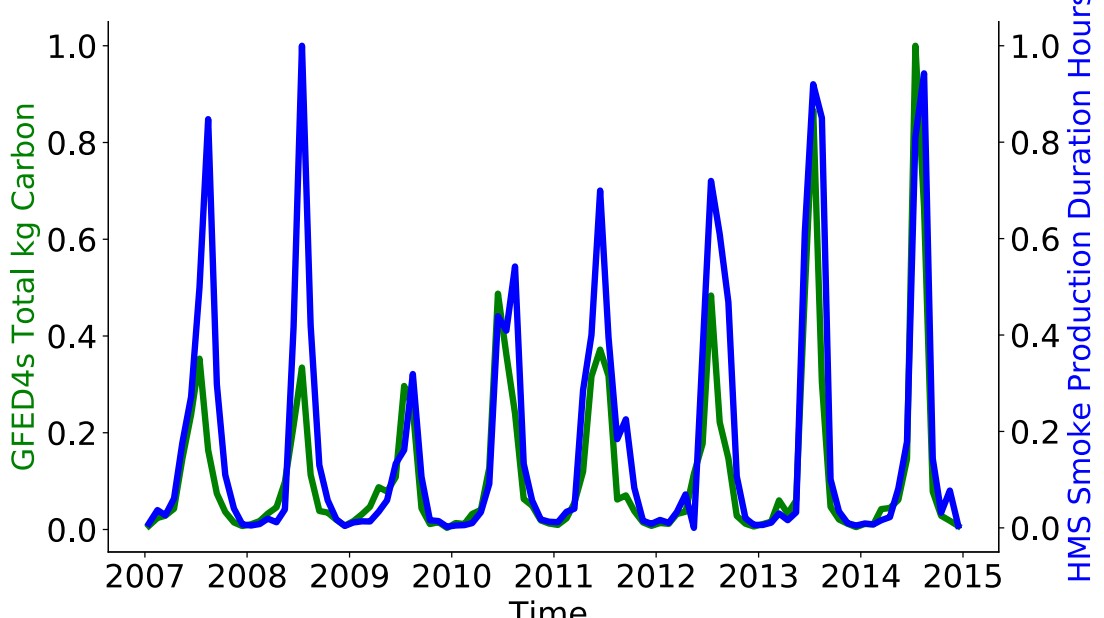

**Figure 4: Time series of monthly domain (20-80$^o$ N, 50-170$^o$ W) summed kg carbon (left axis) and HMS-hours (right axis). Monthly totals have been divided by the maximum monthly value observed between 2007-2014. The Pearson correlation coefficient for the presented monthly time series is 0.842.**







**Figure 5: Plan view monthly time series Pearson correlation coefficient for monthly HMS-hours and GFED4s kg carbon for each 0.25 x 0.25 degree grid cell. White grid cells are locations where one or both of the inventories have zero emissions for all 96 months.**




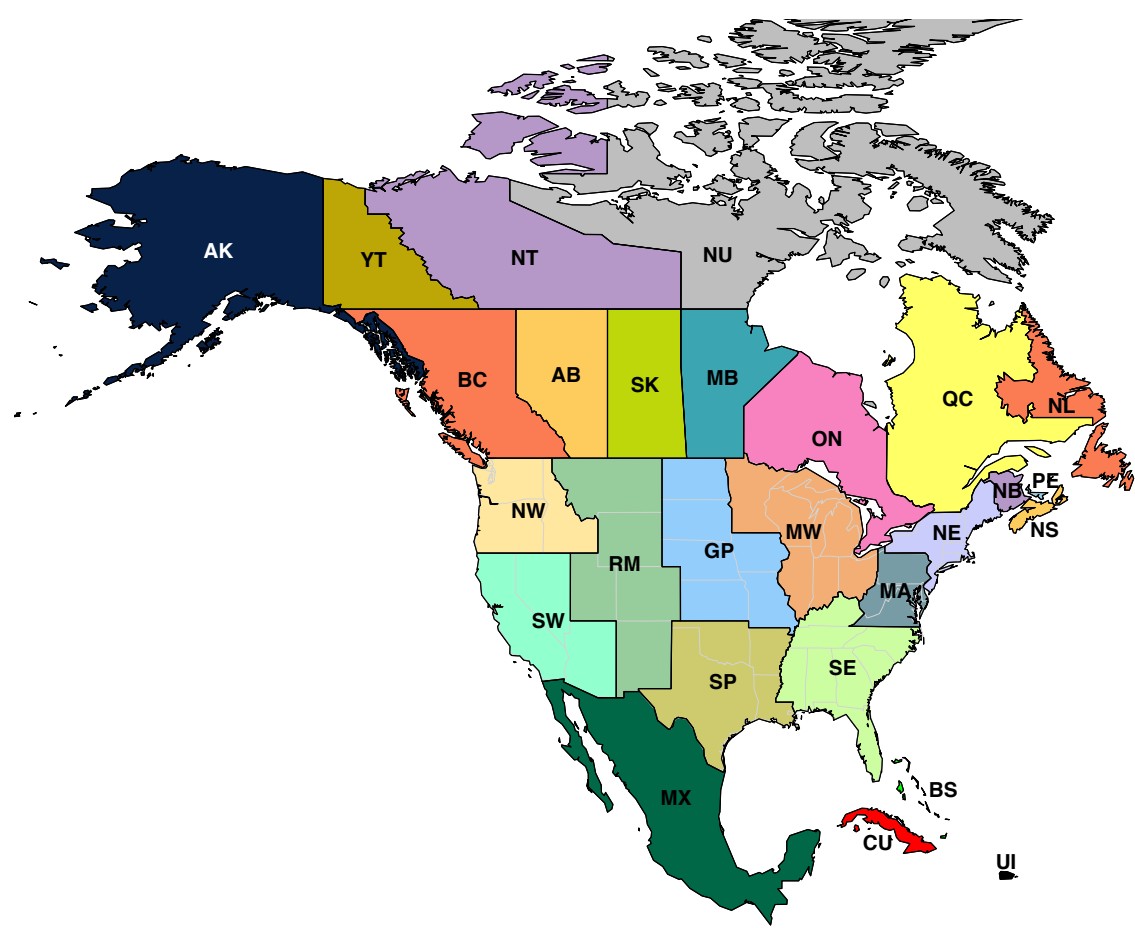

**Figure 6: Smoke source and receptor regions used in this analysis. Northeast (NE), Mid Atlantic (MA), Southeast (SE), Midwest (MW), Southern Plains (SP), Great Plains (GP), Rocky Mountains (RM), Southwest (SW), Northwest (NW), Alaska (AK), U.S. Islands (UI), Mexico (MX), Quebec (QC), Nova Scotia (NS), Saskatchewan (SK), Alberta (AB), Newfoundland and Labrador (NL),**
5 **British Columbia (BC), New Brunswick (NB), Prince Edward Island (PE), Yukon Territory (YT), Manitoba (MB), Ontario (ON), Nunavut (NU), Northwest Territories (NT), Cuba (CU), and Bahamas (BS).**




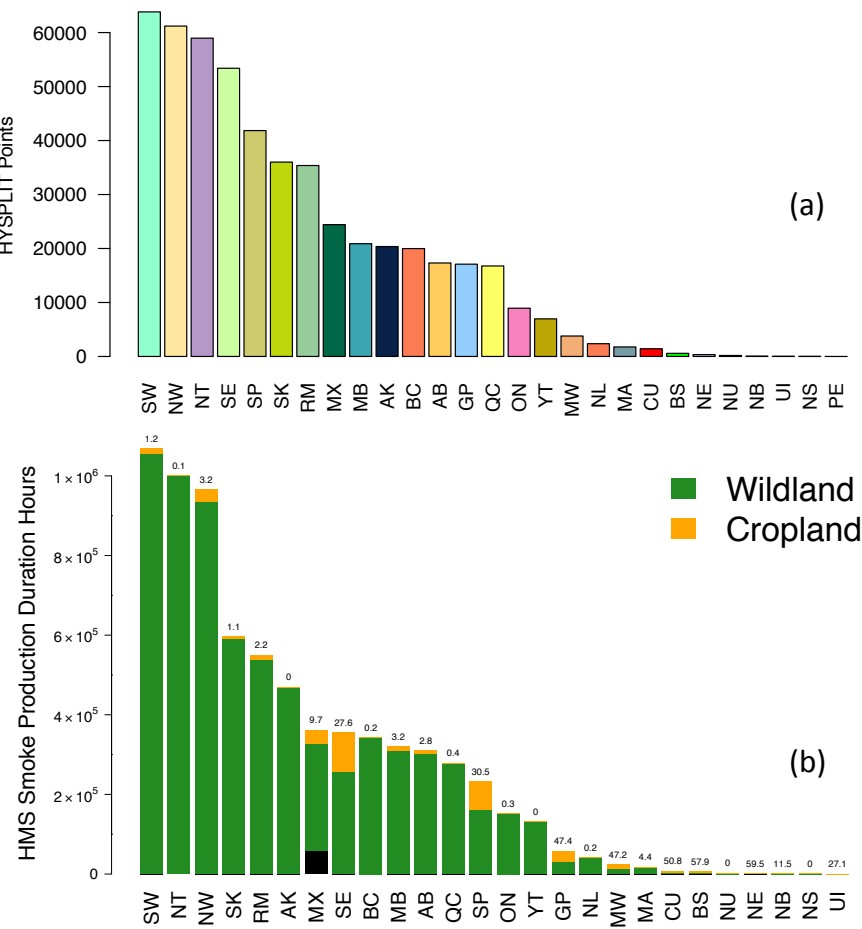

**Figure 7: (a) Total HMS HYSPLIT point detections for all regions and months between 2007-2014. Region colors and abbreviations are defined in Fig. 6. (b) Total HMS smoke production duration hours (SPDH) for the same time period. The**
5 **number on top of each bar indicates the percentage of SPDH that are from croplands. The black portion of Mexico (MX) represents fires that occurred on land outside of the land cover data extent used in this analysis.**





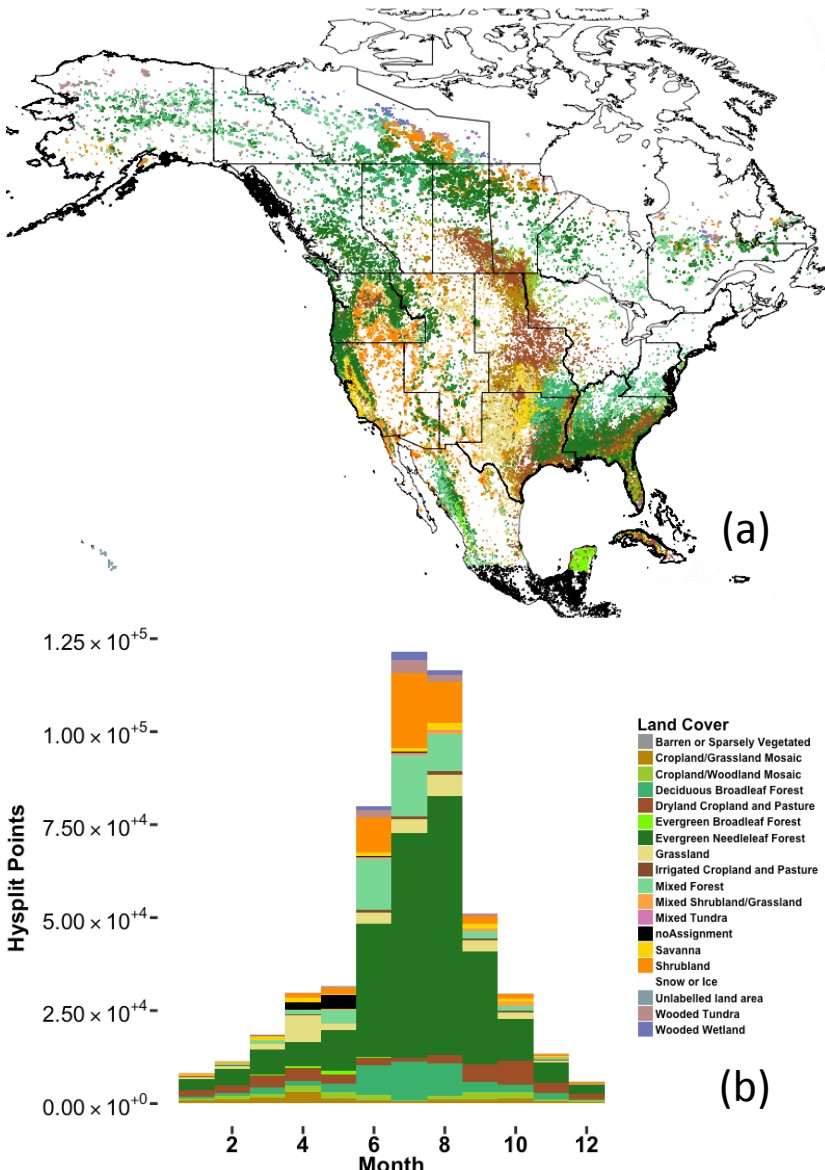

**Figure 8: Location, (a) seasonality, (b) and land cover classification assignment (color) for all HYSPLIT points analyzed over North America between 2007-2014 (n=517,214). No land cover assignment is made for Southern Mexico (MX) and U.S. Islands**
5 **(UI) due to the latitudinal range of land cover data.**





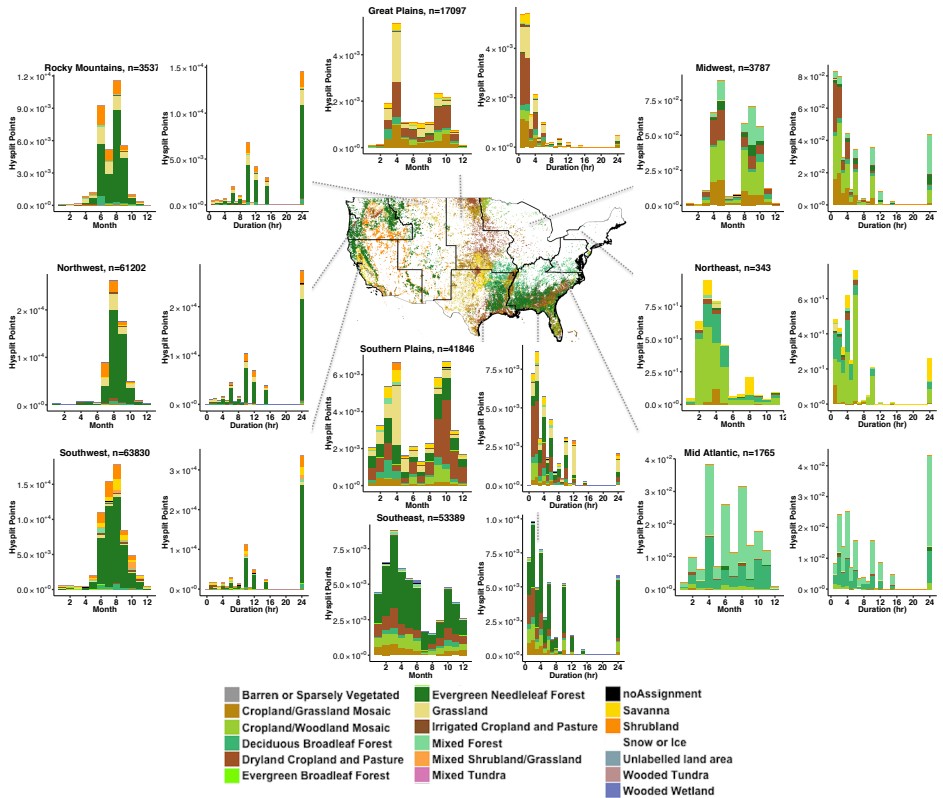

**Figure 9: HYSPLIT point locations and land cover classification for the nine continental U.S. regions. The map in the middle shows the regions and the locations of all HYSPLIT points analyzed between 2007 and 2014. The HYSPLIT points are shaded by land cover classification. Two histograms are shown for each region. The total number of HYSPLIT points analyzed each month is shown on the left histograms. The HYSPLIT point duration and land cover classification is shown on the right. The bars for both histograms are shaded by the land cover classification assigned to the HYSPLIT points. Non U.S. regions and Alaska are available in the supplemental information.**





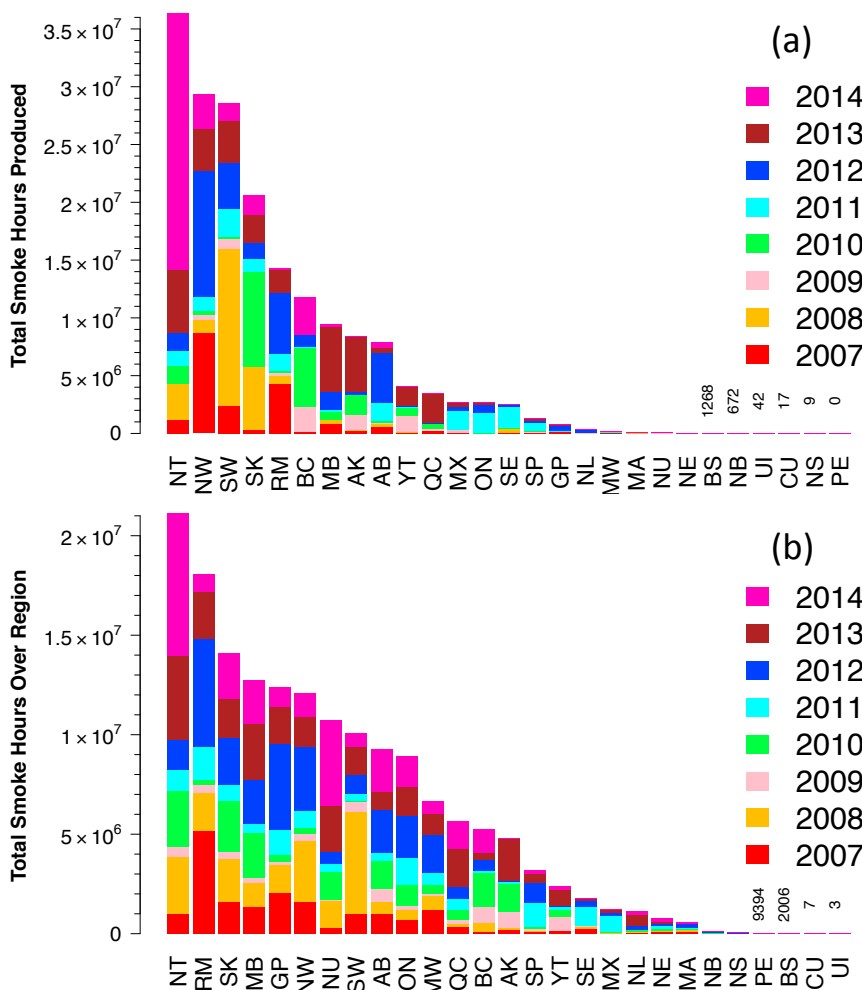

Figure 10: (a) Total number of smoke hours produced (anywhere) by each region for months June-September between 2007 and 2014 using GDAS1 meteorology. (b) Total number of smoke hours over each region for months June-September between 2007 and 2014 using GDAS1 meteorology.





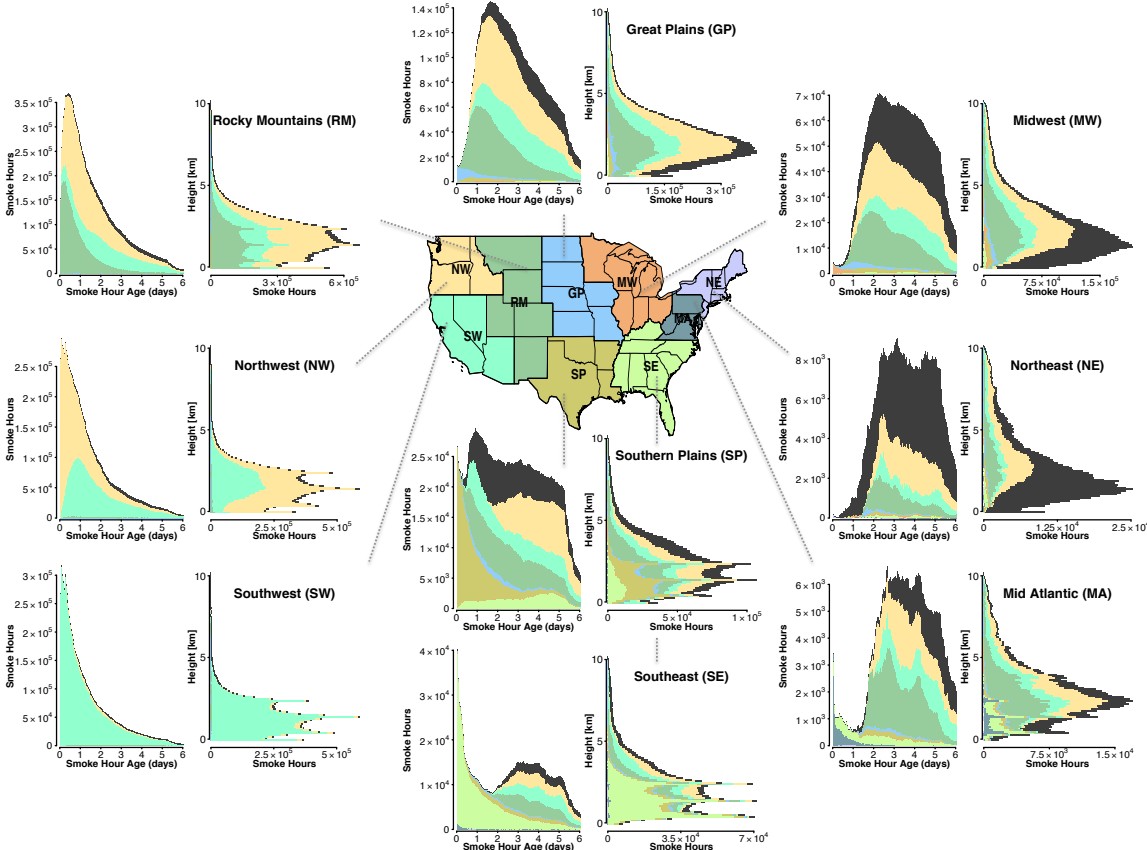

**Figure 11: Summertime (June-September) regional smoke hour transport summaries for the nine continental U.S. regions between 2007 and 2014 using GDAS1 meteorology. Two histograms are shown for each region. The histogram on the left shows the distribution of smoke hour age. The color of the smoke hours matches the region of origin as indicated by the map in the center. Column 2 shows the distribution of smoke hour height segregated by source region color. Smoke hours contributed by regions outside of the continental U.S. are dark gray. Histograms of smoke hour age, height, and region of origin for all regions shown in Fig. 6 are available in the supplemental information.**



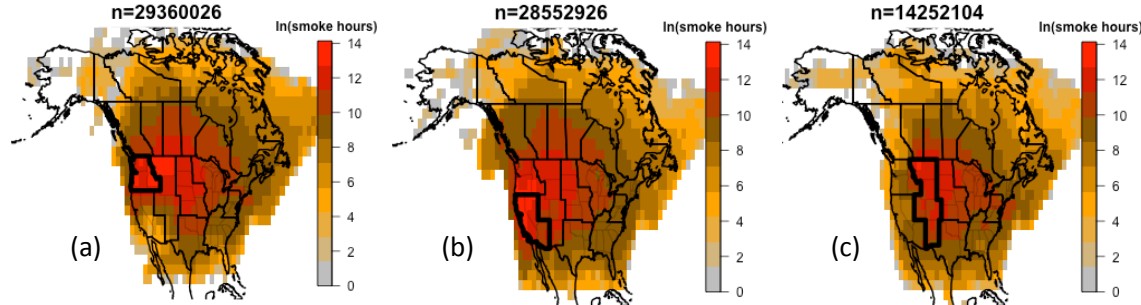

**Figure 12: Summertime (June-September) regional smoke hour transport heat maps for the Northwest (a), Southwest (b), and Rocky Mountains (c). Each column shows the count of smoke hours produced by each region on a 2° by 2° degree grid with a consistent color bar for all three regions. The grid spans 18-180°W and 18-90°N, a domain covering all five sectors where HMS analyzes smoke plumes (only a subset plotted). Shaded values are the natural log of the number of smoke hours in each grid cell (min=1, max=1.2 million). All figures were generated using GDAS1 meteorology data.**

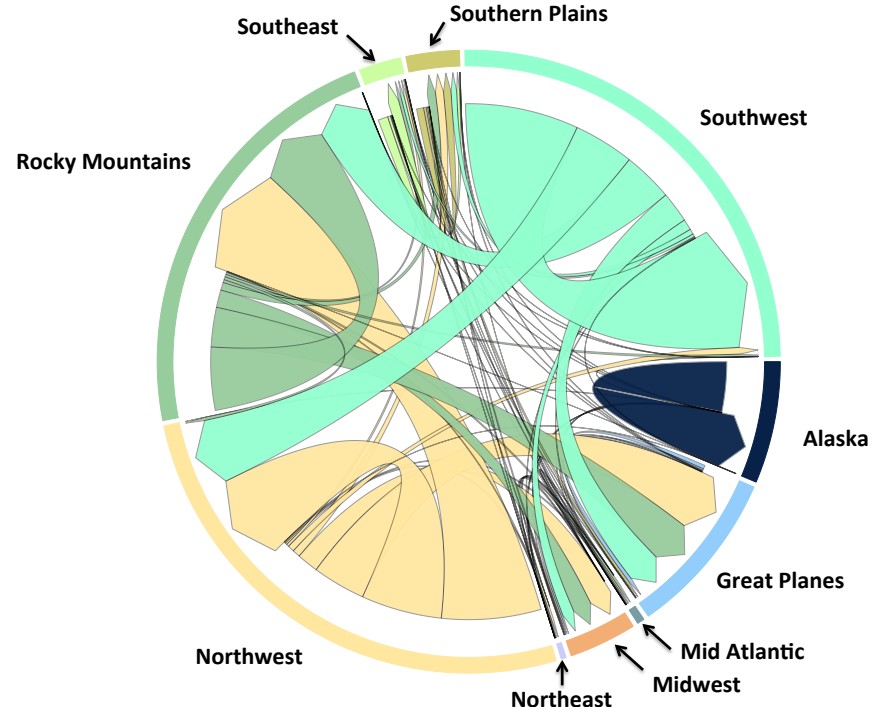

**Figure 13: U.S. regions smoke hour climatology for June-September 2007-2014 using GDAS1 meteorology. Arrows indicate the smoke hours source region and destination region. The width of arrows are proportional to total smoke hours transported between regions. Each of the 10 U.S. regions makes up a fraction of the total circumference of the circle, which is proportional to the total smoke hours over and produced by all U.S. regions. The widths of the arrows are proportional to the number of smoke hours in a given transport pathway. The colors of the arrows match the source region of the smoke hours (Fig. 6). This figure places the individual histogram of Fig. 11 into context.**