# Peer review of "Connecting smoke plumes to sources using Hazard Mapping System (HMS) smoke and fire location data over North America"

_Atmospheric Chemistry and Physics, 2017_

## Referee Comment (RC1) · Anonymous Referee #2 · 13 Jul 2017

Dear Authors,

Thank you for a well-written concise manuscript describing your very interesting experiment. Leveraging the thousands of hours of analyst labor manifest in the NOAA hazard mapping system for science purposes is a very worthy goal. The basic climatological analysis of smoke influences over the US could not readily be performed without these HMS data.

I consider your study worthy of publication, but your results are only semi-quantitative and in some cases potentially subject to large errors, because of weaknesses in the input datasets. You will need to at least include a discussion of these potential errors

and hopefully some analysis to approximate their magnitude in your final paper.

Land cover is the most egregious example: while the US landscape has not been radically reshaped over the past 25 years, this is not an excuse to use a truncated version of a dataset based on 25-year-old AVHRR data. The Hansen et al. 2000 paper with basic validation results for this dataset is a good place to start, it says "Comparisons of the final product with regional digital land cover maps derived from high-resolution remotely sensed data reveal general agreement, except for apparently poor depictions of temperate pastures within areas of agriculture" (http://www.tandfonline.com/doi/abs/10.1080/014311600210209).

I do not think that your results would see large qualitative changes if you used a more modern dataset such as the North American Land Change Monitoring System (https://landcover.usgs.gov/nalcms.php) but I would expect much better answers from that dataset in areas such as the discrimination of cropland and forested land in the Southeast US (the best dataset for that purpose would be the Cropland Data Layer [https://nassgeodata.gmu.edu/CropScape/] ). I am not recommending that you redo your entire analysis with a different land cover dataset (though it might not be that difficult to do so). However, you should include this in your discussion of uncertainties.

You are also using the HMS analyst-generated fire detection data, and for those data there is a published validation: Schroeder et al. IJRS 2008 (http://www.tandfonline.com/doi/abs/10.1080/01431160802235845) . This paper indicates good quality of the fire data, and does not point to significant source of error except to note that like all fire detection systems, small fires are much harder to detect and will be systematically underrepresented in the output products. A recent paper by Hu et al. JGR 2016 (http://onlinelibrary.wiley.com/doi/10.1002/2015JD024448/abstract) describes how these errors manifest as both drastic underdetection of individual fires and as imbalances in fire detection rates by ecosystem. The agreement in the coarse seasonality of North American burning between HMS and GFED data, while encouraging, does not rule out significant biases at the scale of your regional analysis.

You cite the Rolph et al. (2007) paper about the NOAA Smoke Forecasting system, but you need to include the information on uncertainties from that paper in your discussion. That paper found very weak agreement between HMS smoke plumes and smoke transport model results, and while that paper was formulated as a validation of the transport model using HMS plumes, it remains that there is no published validation of the HMS smoke plume extent data, and it is likely to have both large uncertainties as well as some systematic biases due to discrimination of smoke being easier over some areas and seasons relative to others. The Rolph paper is a good place to start to formulate a discussion of how potential errors in the HMS smoke extent data could affect your results.

This last area is one where I will recommend additional analysis. Your current manuscript includes this analysis (page 2): "10% of these plume days are days where ground level PM2.5 is one standard deviation above average summertime concentrations." That 10% was for Minnesota stations; you cite a figure of 30% for Washington and Oregon stations. This is a good basis for a test of the skill of your method; however, because you are using ground monitors, the additional uncertainty of the vertical profile means that no conclusion can be drawn from these results. I recommend repeating this analysis using AERONET stations in the Western and Eastern US, to determine whether the presence of HMS-diagnosed smoke corresponds with significantly elevated aerosol optical depth relative to the seasonal mean values. This analysis would build confidence in the unvalidated HMS smoke plume extent that is the core of your study.

Good luck with completion of this study, and I look forward to its publication, but I hope to see an expanded discussion of the uncertainties in your analysis that will assist readers in drawing conclusions from these unique comprehensive datasets.

---

## Referee Comment (RC2) · Anonymous Referee #1 · 8 Sep 2017

This manuscript primarily details an attribution analysis of the relative contribution of biomass burning smoke originating from the North America region to smoke observed over predefined areas within the continental US. The authors employ NESDIS HMS data and forward trajectory modelling using HYSPLIT to achieve their analysis and results. A smoke transport climatology is presented, which outlines the key smoke producing regions and their influence over themselves and other neighbouring receptor regions.

The narrative is generally well written and logically organised, with clear figures and diagrams – particularly the visual analytics style graphic at the end which provides a

nice overview of the smoke pathways. The analysis of the smoke climatology follows a reason approach and the results are presented clearly.

In my opinion the main deficiencies are in the input datasets used for the analysis as outlined in the comments below, and a lack of discussion on the potential margin of error in the results. Overall, I think the results are of interest to the community, and would recommend publication after the following comments have been addressed.

1. The HMS is used operationally for smoke forecasting but clarification is needed why this is considered a more suitable choice of dataset for this paper over other established ones like GFED for example, which is compared in the paper and widely used in many studies. Since the HMS product is considerably subjective as it is based on analysts manually adding points for various situations as outlined in section 2.1, the consistency of the product needs to be put into question as a suitable dataset for such analysis. There are also limitations on available years of suitable HMS data. The subjectivity and inconsistency of this operational dataset also limits its usefulness for future analysis.

2. The smoke plume analysis done operationally by the HMS analysts also have a large element of subjectivity and it would be useful to cross check this with another dataset (as was done with the comparison between the HYSPLIT points and GFED). One possibility would be AOD for example, from satellite observations as well as AERONET stations.

3. Units of time (i.e. hours of smoke) are used for the analysis though it would have been better to use derived smoke emissions instead which would take into account land cover characteristics, fuel loading etc. Just using duration alone seems to be a self-imposed limitation when comparing with the amount of smoke observed. Some explanation to better justify this approach would help.

4. The land cover map using data from 1992-93 is considerably old and it is difficult to see why a more updated map wasn't used since there are various newer maps available out there. Unless it could be shown that there weren't significant changes

in the land cover over the 15 years or more to the analysis years of 2007-14 (quite unlikely), the results involving land cover classification are hampered by using the old map dataset.

Specific comments:

1. P1, line 16 & P11, line 30: . . .'HYSPLIT'. . .

2. P2, line 26 -35: There have been attribution studies conducted in other parts of the world using Lagrangian models with chemistry. Emissions information are available from global inventories including GFED, GFAS, FINN for example, so I would disagree that a modelling approach is unsuitable, because it would clearly be more comprehensive and could include full plume dispersion (compared to trajectories), deposition, attribution of secondary smoke particulates etc.

3. P3, line 3: What is meant to "trigger" a smoke forecast?

4. P3, line 26-29: Please clarify how the accuracy statistics stated here were determined.

5. P3, line 30-31: Repeated phrase - "HYSPLIT points in proportion . . . smoke observed".

6. P4, line 5: On the relationship between HYSPLIT points and smoke quantity – is this purely based on duration assigned by the analyst or is land cover taken into account? Do all HYSPLIT points emit the same amount of smoke? This also raises the question again of why hours of smoke are being used instead of derived emissions.

7. P7, line 6: Is the difference in magnitude actually due to comparison between 'SPDH' hours and actual C emissions rather than "varying emission factors for different ecosystems"?

8. P9, line 30-31: Just a comment that a modelling approach would better allow altitude specific analysis to be conducted e.g. at surface level where air quality is of concern to

the population.

9. P11, line 2: It would be good to provide some explanation on why the trajectories run using the EDAS data are nearly identical and if this is something expected given the higher resolution of the meteorological input.

---

## Author Comment (AC1) · 20 Oct 2017

Thanks to the two reviewers for their thoughtful comments on the manuscript. We were able to address all the comments. Please see the attached PDF that contains: 1) Reviewer comments and our responses. 2) The manuscript with track changes indicating changes that were made during the review process (starts on page 16). 3) The updated supplemental information containing new figures and analysis regarding land cover (starts on page 50).

Thanks!

[Figure]

Please also note the supplement to this comment:
https://www.atmos-chem-phys-discuss.net/acp-2017-245/acp-2017-245-AC1-supplement.pdf
* * *

---

## Author Response (AR1)

This document will be where Steven Brey and co-authors outline responses to reviewers of manuscript ACP-2017-245. To make this process easier, we have pasted the text of the reviews into this document and provide responses to reviewer comments immediately after their text.

After the reviewer comments and discussion, we have placed the entire manuscript and show track changes so that it will be easy to see what changes were made to the manuscript as a result of the reviewer comments/discussion. This starts on page 16.

Reviewer Comments are in black text. Our responses to the reviewer have been inserted in blue text. Red text is text that has been added to the manuscript as well as presented here to the reviewer.

Thanks to the two reviewers for their thoughtful comments on the manuscript. We were able to address all the comments. In particular, we took the time to explore the choice of land cover classification. As suggested by the reviewer, this did not change our results meaningfully. However, we have added a substantial number of figures to the Supplemental Materials so anyone interested in using the datasets we present can easily explore this.
* * *
Reviewer 1 Comments:
* * *
Dear Authors,

Thank you for a well-written concise manuscript describing your very interesting experiment. Leveraging the thousands of hours of analyst labor manifest in the NOAA hazard mapping system for science purposes is a very worthy goal. The basic climatological analysis of smoke influences over the US could not readily be performed without these HMS data.

I consider your study worthy of publication, but your results are only semi-quantitative and in some cases potentially subject to large errors, because of weaknesses in the input datasets. You will need to at least include a discussion of these potential errors and hopefully some analysis to approximate their magnitude in your final paper.

Land cover is the most egregious example: while the US landscape has not been radically reshaped over the past 25 years, this is not an excuse to use a truncated version of a dataset based on 25-year-old AVHRR data. The Hansen et al. 2000 paper with basic validation results for this dataset is a good place to start, it says "Comparisons of the final product with regional digital

land cover maps derived from high-resolution remotely sensed data reveal general agreement, except for apparently poor depictions of temperate pastures within areas of agriculture" (http://www.tandfonline.com/doi/abs/10.1080/014311600210209).

I do not think that your results would see large qualitative changes if you used a more modern dataset such as the North American Land Change Monitoring System (https://landcover.usgs.gov/nalcms.php) but I would expect much better answers from that dataset in areas such as the discrimination of cropland and forested land in the Southeast US (the best dataset for that purpose would be the Cropland Data Layer [https://nassgeodata.gmu.edu/CropScape/] ). I am not recommending that you redo your entire analysis with a different land cover dataset (though it might not be that difficult to do so). However, you should include this in your discussion of uncertainties.

Our original choice in the use of the 1992 National Center for Earth Resources Observation and Science land cover dataset was driven by the availability of the dataset in an easy to use gridded format that was consistent over all North America. The dataset is available in a non proprietary format. However, we agree with the reviewer that using a different, updated land cover dataset is an important test of the uncertainty associated with this choice. Therefore, following the reviewer's recommendation, we have taken the time to assign HYSPLIT points land cover using a dataset from the suggested *North American Land Cover Change Monitoring System.* The specific dataset we choose to use within this system was the *2010 Land Cover of North America at 250 meters* data published in 2013 by the Commission for Environmental Cooperation in Montréal, Québec, Canada (http://www.cec.org/tools-and-resources/map-files/land-cover-2010). These data are based on the Moderate Resolution Imaging Spectroradiometer (MODIS/TERRA) seven land spectral bands top of atmosphere reflectance. These data were created primarily to assess North American land cover changes between 2005 and 2010. Approximately 1% of land area land cover classifications changed during this time period.

Overall we find that the differences between the two datasets are driven by an inconsistent list of land cover classifications between the 1992 and 2010 data. Comparing the two different land cover classifications is challenging and requires subjectivity due to the fact that the land cover classification categories are not the same between the 1992 and 2010 data. For example, when assessing whether the two datasets agree on land cover for a given location there needs to be a decision as to whether "temperate or subpolar needleleaf forest" and "evergreen needleleaf forest" count as a match. As indicated by the reviewer, making this decision is not particularly pertinent to the presentation of land cover dependent results presented in the paper. Thus we did not re-do the figures in the main body of the text using the alternative land cover dataset. Instead we added figures to the Supplemental Information (S9.1 Fig. S8 - S24) materials that show how the HYSPLIT point land cover designations differ between the two datasets. We have sprinkled this throughout the text where HYSPLIT land cover is discussed.

Additions:

- Section 2.3
  - "We point readers to Hansen et al., 2000 for a discussion of the weaknesses of the 1992 land cover dataset. We have also completed a comparison to a 2010 land cover dataset, and this comparison is available in the Supplemental Information Sect 9."
- Section 4, paragraph 1
  - "When we assess the land cover classification of HYSPLIT points using a 2010 land cover dataset, the most common alternative assignment for the most abundant cropland assignment type ("cropland/grassland mosaic") is "temperate subpolar grassland". Regional differences for alternative assignments exist. For example, in the U.S. Midwest and Great Plains, the most common alternative land cover assignment for cropland is "temperate subpolar grassland" while in the Southeast U.S. the most common alternatives are "temperate subpolar needleleaf forests" and "wetland". These 2010 data show that the most common alternative land cover assignment for forested lands in North America is "subpolar shrubland"."

Citation for 2010 land cover data used in the supplemental material:

2005 North American Land Cover at 250 m spatial resolution. Produced by Natural Resources Canada/Canadian Center for Remote Sensing (NRCan/CCRS), United States Geological Survey (USGS); Insituto Nacional de Estadística y Geografía (INEGI), Comisión Nacional para el Conocimiento y Uso de la Biodiversidad (CONABIO) and Comisión Nacional Forestal (CONAFOR).

You are also using the HMS analyst-generated fire detection data, and for those data there is a published validation: Schroeder et al. IJRS 2008 (http://www.tandfonline.com/doi/abs/10.1080/01431160802235845) . This paper indicates good quality of the fire data, and does not point to significant source of error except to note that like all fire detection systems, small fires are much harder to detect and will be systematically underrepresented in the output products. A recent paper by Hu et al. JGR 2016 (http://onlinelibrary.wiley.com/doi/10.1002/2015JD024448/abstract) describes how these errors manifest as both drastic underdetection of individual fires and as imbalances in fire detection rates by ecosystem. The agreement in the coarse seasonality of North American burning between HMS and GFED data, while encouraging, does not rule out significant biases at the scale of your regional analysis.

Thank you for pointing out these papers. However, the Hu et al. (2016) paper analyzes the so-called 'hybrid HMS', referring to the fact that the HMS uses multiple satellites (there is only one HMS). The Hu et al. (2016) paper analyzes the automatic detections of the HMS, not the HYSPLIT points presented in this manuscript, thus the detection rate results presented are not applicable to our study. Mark Ruminski is a co-author on our work, and on the Hu et al. (2016) JGR paper. We have included citations to both these papers and we have added an additional sentence reminding readers that any satellite-based detection system (whether automatic or human vetted) is likely to struggle with the detection of small fires and be limited to low cloud optical depth conditions.

We have added the following to the manuscript:

"Schroeder et al. (2008) assess the automatic detections; there are small false alarm rates (~2%) the conterminous U.S. As in all satellite-based fire detection systems, small fires are more difficult to detect and are under-reported (*e.g.* Hu et al., 2016)."

You cite the Rolph et al. (2007) paper about the NOAA Smoke Forecasting system, but you need to include the information on uncertainties from that paper in your discussion. That paper found very weak agreement between HMS smoke plumes and smoke transport model results, and while that paper was formulated as a validation of the transport model using HMS plumes, it remains that there is no published validation of the HMS smoke plume extent data, and it is likely to have both large uncertainties as well as some systematic biases due to discrimination of smoke being easier over some areas and seasons relative to others. The Rolph paper is a good place to start to formulate a discussion of how potential errors in the HMS smoke extent data could affect your results.

We had already included this discussion in the methods (Section 2.1 and 2.2), but now we note also that there is no comprehensive validation of the HMS plumes. We are interested in pursuing a validation of this dataset, but it is certainly beyond the scope of this paper. Potentially useful datasets include AERONET (as mentioned later by both reviewers), and also CALIPSO/CATS. However, validation will need to be thoughtfully designed to test the known specific weaknesses. For example, the largest uncertainty is likely to be at the edges of the plumes. We agree with the reviewer that it is ideal to remind readers of these caveats in the discussion and conclusions. So, we have used that approach here.

The following text is now included in the updated Section 2.2:

"Rolph et al. (2009) used the HMS smoke plumes to validate NOAA's operational smoke forecasting system. There is no published validation of the HMS smoke plume extent data. It could have systematic biases because discrimination of smoke is easier over some areas, seasons,

and under certain synoptic weather conditions. For our aggregate smoke plume analysis, the largest uncertainty is at the edges of the plumes. The temporal information is also a source of uncertainty. During the day, the extent can be assessed hourly. However, this is not possible overnight. Smoke is sometimes transported to areas with anthropogenic haze pollution. In some cases the smoke will mix with and become indistinguishable from the anthropogenic haze pollution. The greater the distance travelled by a smoke plume, the more challenging it is to distinguish between smoke and anthropogenic haze. This challenge is particularly pronounced for aged smoke impacting the Southeastern U.S."

The following has been added to section S10 (*Additional HMS smoke plume operational analysis details* section of the supplemental material)

"Presently, there is no comprehensive validation of HMS smoke plume analysis. One of the reasons in that there is no spatially and temporally comprehensive ground truth to compare to. Other satellite data would need to be used and these would have their own uncertainties. In the manuscript we state that the largest uncertainty for the smoke plume analysis is likely to be the edges of the smoke plume. In aggregate this is true, but may not be the largest uncertainty for every individual smoke plume."

We have updated the Discussion and conclusions with the following sentences:

- Section 6, paragraph 1
    - "The location of smoke hours is restricted to locations where smoke plumes have been analyzed while the height is estimated from HYSPLIT forward trajectories."
- Section 6, paragraph 1
    - "This dataset confirms that there is a substantial amount of smoke-producing fire activity in the Southeast, particularly along the lower Mississippi River Valley. This is likely an underestimate of the contribution of small fires to smoke abundance in this region given that small fires within this region can be substantially under-detected (Hu et al., 2016). In addition, it can be challenging to differentiate anthropogenic haze from smoke in this region."

This last area is one where I will recommend additional analysis. Your current manuscript includes this analysis (page 2): "10% of these plume days are days where ground level PM2.5 is one standard deviation above average summertime concentrations." That 10% was for Minnesota stations; you cite a figure of 30% for Washington and Oregon stations. This is a good basis for a test of the skill of your method; however, because you are using ground monitors, the additional uncertainty of the vertical profile means that no conclusion can be drawn from these results. I recommend repeating this analysis using AERONET stations in the Western and Eastern US, to determine whether the presence of HMS-diagnosed smoke corresponds with significantly

elevated aerosol optical depth relative to the seasonal mean values. This analysis would build confidence in the unvalidated HMS smoke plume extent that is the core of your study.

This is a miscommunication. We are not attempting to validate the HMS with the ground sites, and Figure 1 is presented before we even discuss the plumes themselves. Our intention was for this section to demonstrate that plumes do not necessarily result in elevated $PM_{2.5}$ values. This is an important piece of information to properly interpret the rest of the paper. An HMS smoke plume analyzed overhead does not ensure ground level air quality impacts. Figure 1 presents a summer climatology of the percent of time an overhead smoke plume has resulted in elevated ground level $PM_{2.5}$. When we point out that 10% of so-called plume days in Minnesota are observed to have elevated surface $PM_{2.5}$ concentrations we are not validating the HMS, we are pointing out that most plume days in Minnesota do not result elevated ground level $PM_{2.5}$. We have added the following sentences to this section to make this clear to all readers.

"Fig. 1 is not intended to validate the HMS smoke product. This figure is meant to set the stage for interpreting the analysis presented in later sections, *i.e.* a smoke plume overhead does not necessarily ensure elevated ground-level $PM_{2.5}$ concentrations."

The following sentence has been added to the conclusion section:

"The location of smoke hours is restricted to locations where HMS analysts have drawn smoke plumes while the height is estimated from HYSPLIT forward trajectories emanating from HYSPLIT points."

Good luck with completion of this study, and I look forward to its publication, but I hope to see an expanded discussion of the uncertainties in your analysis that will assist readers in drawing conclusions from these unique comprehensive datasets.

We thank you for your thoughtful review! We also want to provide the reviewers, editor, and readers with the following online interactive data viewing tools, which show raw HMS data. This data visualization increases the transparency of this work by showing the reader exactly how this data looks on a daily timescale.
- http://sjbrey.atmos.colostate.edu/HMSExplorer/
- http://sjbrey.atmos.colostate.edu/smokeWheel/ (interactive version of Figure 13)
* * *
Reviewer Comments are in black text. Our responses the reviewer have been inserted in blue text. Red text is text that has been added to the manuscript as well as presented here to the reviewer.
* * *
This manuscript primarily details an attribution analysis of the relative contribution of biomass burning smoke originating from the North America region to smoke observed over predefined areas within the continental US. The authors employ NESDIS HMS data and forward trajectory modelling using HYSPLIT to achieve their analysis and results. A smoke transport climatology is presented, which outlines the key smoke producing regions and their influence over themselves and other neighbouring receptor regions.

The narrative is generally well written and logically organised, with clear figures and diagrams – particularly the visual analytics style graphic at the end which provides a nice overview of the smoke pathways. The analysis of the smoke climatology follows a reason approach and the results are presented clearly.

Thank you for your feedback of our final figure! There is an interactive version of Figure 13 available at the following url: http://sjbrey.atmos.colostate.edu/smokeWheel/

In my opinion the main deficiencies are in the input datasets used for the analysis as outlined in the comments below, and a lack of discussion on the potential margin of error in the results. Overall, I think the results are of interest to the community, and would recommend publication after the following comments have been addressed.

Thank you. In response to Reviewer 1, we have added discussion on uncertainty resulting from:
- HMS HYSPLIT points (section 2.1)
- HMS smoke plumes (section 2.2)
- Section 5 & 6 (discussion and conclusion)

Please see specific responses and additions below.

1. The HMS is used operationally for smoke forecasting but clarification is needed why this is considered a more suitable choice of dataset for this paper over other established ones like GFED for example, which is compared in the paper and widely used in many studies. Since the HMS product is considerably subjective as it is based on analysts manually adding points for various situations as outlined in section 2.1, the consistency of the product needs to be put into question as a suitable dataset for such analysis. There are also limitations on available years of suitable

HMS data. The subjectivity and inconsistency of this operational dataset also limits its usefulness for future analysis.

The HMS human generated fire detections (HYSPLIT points) and smoke plumes are particularly valuable from an operational air quality perspective. Fire emission inventories, such as GFED, are useful to initialize model simulations. However, those model simulations are subject to additional uncertainties. For example, injection height is very difficult to simulate properly. In addition, GFED does not have a smoke plume product created in-tandem with the fire locations the way the HMS system does. Because this work is concerned with where smoke plumes originate and travel, this component of the work would not be possible using GFED or another "traditional" emission inventory. Case-study validation of the ability of chemical transport models (*e.g.* GEOS-Chem, WRF-Chem, *etc.*) shows that these models have difficulty 1) accurately representing smoke plume spatial extent, 2) cannot be run at 1 degree grid spacing at the daily or hourly timescale for the length of time needed for this analysis (*i.e.* covering 8 years) due to computational limitations. GFED also does not provide sub-daily information on fire occurrence the way HYSPLIT points do (hourly), this could have consequence for the timing of emissions and transport pathways. Though there are other emission inventories that have a higher time resolution *(e.g.* FINN), similar modelling challenges still result when this inventory is implemented into chemical transport models.

Gridded emission inventories do not provide the same spatial or temporal information as the HMS HYSPLIT points. GFED4s arguably represents the state of the science when it comes to emission inventories. This product is gridded to a $0.25 \times 0.25^{o}$ (~27 x 27 km) grid, which is much more coarse than the ~2-3 km accuracy of individual HYSPLIT points.

This paper offers orthogonal methods to answer smoke transport questions that are usually answered using gridded emission inventories and chemical transport models. Understanding the impact of smoke on the U.S. airshed is a very important problem that should be investigated using multiple methods. Because HMS HYSPLIT points are the analysed fires used to initialize US National Weather Service Air Quality Forecasts they represent a unique subset of fires, those that have been determined to potentially be of importance to air quality.

When developing or presenting a new dataset, due diligence requires comparing it to existing datasets. That is our goal in presenting a comparison with GFED4s. We also use this comparison to understand where approaching the problem of smoke transport with different underlying datasets/techniques may lead to different results.

2. The smoke plume analysis done operationally by the HMS analysts also have a large element of subjectivity and it would be useful to cross check this with another dataset (as was done with

the comparison between the HYSPLIT points and GFED). One possibility would be AOD for example, from satellite observations as well as AERONET stations.

In response to Reviewer 1, we had already updated our discussion of uncertainty in the methods (Section 2.1 and 2.2). We specifically note that there is no comprehensive validation of the HMS plumes. We are interested in pursuing a validation of this dataset, but it is certainly beyond the scope of this paper. Potentially useful datasets include AERONET (as mentioned later by both reviewers), and also CALIPSO/CATS. Validation will need to be thoughtfully designed to test the known specific weaknesses, rather than simply comparing AOD across the entire time period or geographic extent of this analysis. The largest uncertainty is likely to be at the edges of the plumes. This is likely what would be best to validate. However, we remind readers that the HMS smoke product is actually used to validated plume models (*e.g.* Rolph et al., 2009) as was mentioned by Reviewer 1.

The strengths and limitations of the smoke plume analysis are accurately described in section 2.2 and that the strengths justify the work and results presented in this paper. We have also added more discussion of the limitations of this work to both the discussion and conclusions sections.

3. Units of time (i.e. hours of smoke) are used for the analysis though it would have been better to use derived smoke emissions instead which would take into account land cover characteristics, fuel loading etc. Just using duration alone seems to be a self-imposed limitation when comparing with the amount of smoke observed. Some explanation to better justify this approach would help.

The goal of this work is to identify the common transport pathways associated with fires that initiate air quality forecasts. Our goal is not to predict the amount of smoke impacting a given surface site, and in fact, this work may be useful outside the air quality community because we have generically described smoke transport in the column. Smoke has radiative impacts in addition to composition impacts. Moving toward an emission approach would actually increase the uncertainty associated with this analysis. This would essentially scale our results by a set of emission factors, combustion efficiency estimates, fuel load estimates, *etc.*, and would not change the overall findings of the paper. In addition, using hours helps us better understand how quickly smoke plumes are moving and dispersing. It also helps us view smoke age efficiently, which is another goal of this paper.

4. The land cover map using data from 1992-93 is considerably old and it is difficult to see why a more updated map wasn't used since there are various newer maps available out there. Unless it could be shown that there weren't significant changes in the land cover over the 15 years or more to the analysis years of 2007-14 (quite unlikely), the results involving land cover classification are hampered by using the old map dataset.

Refer to the response of reviewer number 1 and our analysis showing just how different these are and if land cover matters. We have added several figures to the supplemental material (S9, S8 - S24) showing differences in land-cover characteristics assigned to HYSPLIT points using a 2010 dataset.

Specific comments:

1. P1, line 16 & P11, line 30: . . .'HYSPLIT'. . .

Thank you for catching these typos. Both instances of 'HYSPIT' have been changed to HYSPLIT.

2. P2, line 26 -35: There have been attribution studies conducted in other parts of the world using Lagrangian models with chemistry. Emissions information are available from global inventories including GFED, GFAS, FINN for example, so I would disagree that a modelling approach is unsuitable, because it would clearly be more comprehensive and could include full plume dispersion (compared to trajectories), deposition, attribution of secondary smoke particulates etc.

We agree that smoke attribution can be accomplished using chemical transport models, and this would allow for additional calculations. However, the strength of Figure 1 is that it is observationally grounded. We also note that our trajectories are only used when validated by observed smoke plumes and that CTMs struggle to accurately simulate smoke dispersion. We have changed this paragraph to read:

"The aggregate view provided by Fig. 1 is unique because it is observationally grounded. There are a number of challenges associated with using a chemical transport model (CTM) to produce an analogous representation of Figure 1. Rastigejev et al. (2010) show that CTMs have difficulty representing plume dispersion, but the plan view result of this process is captured in the HMS smoke plume analysis.  There are other factors contributing to the modelling challenge (*e.g.* Lassman et al.(2017) and references within). For example, there have been a number of advances made in estimating the emissions inputs by improving burned area products (*e.g.* Randerson et al. (2012)) and combining these with emission factors for a wider range of trace species (*e.g.* Akagi et al. (2011) and Wiedinmyer et al. (2011). However, incorporating the full suite of emitted species into models and simulating the rapid chemical evolution of smoke remains a challenge (Alvarado et al., 2015), as is proper treatment of injection height (Paugam et al., 2016) and the timing of emissions (Saide et al., 2015). Models are also subject to uncertainty associated with meteorological inputs (Garcia-Menendez et al., 2013). Finally, running and analyzing a chemical transport model at the fine grid resolution appropriate to simulate all the individual smoke plumes of interest from North American fires over the scale of a decade is currently too computationally expensive to be practical. Thus other lenses are needed to examine how the

smoke from North American fires is transported and dispersed in the atmosphere over seasonal and interannual timescales."

"Due to the difficulty in detecting small fires, it is possible that the smoke hour budget in these regions underestimates local contributions of smoke in these regions."

Added to section 6 where the age of transported smoke plumes is discussed:

"Due to the challenge of distinguishing aged smoke plumes from anthropogenic haze in the Southeast, the abundance of aged smoke hours is likely underestimated in this region."

3. P3, line 3: What is meant to "trigger" a smoke forecast?

Trigger is used here to have the same meaning as "Initiate". The sentence has been changed to "of fires that **initiate** National Weather Service (NWS) smoke forecasts".

4. P3, line 26-29: Please clarify how the accuracy statistics stated here were determined.

The following text has been added to section 2.1 (describing HYSPLIT points)

"GOES visible band and polar orbiting satellites used by the HMS have 1 km resolution at nadir. The 2-3 km accuracy of the location of HYSPLIT points is an estimate based on the additional uncertainties introduced by navigation errors and loss of resolution for observations made away from nadir."

"Visible satellite imagery was available every 30 minutes during the study period, HMS analysts estimate that this makes the start time of smoke emissions accurate to within 1-2 hours."

Mark Ruminski (co-author and HMS team leader) exact comments on this matter:
        "The reviewer is asking about the estimated accuracy in location of HYSPLIT points, in the start time assigned to them and the duration. For the location, this is based on the fact that we use visible imagery to identify smoke plumes. VIS imagery, whether polar or GOES, is at 1km resolution at nadir. By estimating 2-3 km accuracy I tried to account for navigation errors, loss of resolution as you move from nadir (i.e. VIS resolution for GOES-E in the western US is not 1km but more like 2-3 km). etc. For wildfires we also apply HYSPLIT points during the overnight period (no VIS imagery). For this we utilize the 4 micron channel on the polar satellites where the resolution at nadir again is 1 km but decreases as you move toward the limb.
        For the start time, we normally go with the time when we first see smoke from the fire. For a very small percentage of the fires that start overnight (and we can't see smoke) we can add

HYSPLIT points when we see the hotspot. For the study period, we had imagery available every half hour. So we should be able to determine smoke that is visible within that time frame. Finally, for the duration, as is noted in the text, it is more uncertain since many of the ag/prescribe burns continue to produce smoke after usable visible imagery, so we cannot see when the smoke emissions cease. But our assumption is that if they are indeed ag or prescribed, they are not likely to continue burning through the night, so we normally assign a duration that will end an hour or two after sunset. For wildfires, we don't know when they will end. That depends on many other factors. We assume that they will continue to burn until we no longer see smoke one day. On top of all this there is analyst subjectivity. The estimates I provided are based on the above and are my best estimate."

5. P3, line 30-31: Repeated phrase - "HYSPLIT points in proportion . . . smoke observed".

P3, line 30-31 has been rewritten such that there is no repeated phrase. The new wording is shown below.

"Single fires that produce notable amounts of smoke are associated with a cluster of co-located, or nearly co-located, HYSPLIT points assigned roughly in proportion to the amount of smoke observed. The intended operational consequence of designating HYSPLIT points this way is to allow the NWS smoke forecast model to generate more smoke for large fires and less smoke for smaller fires."

6. P4, line 5: On the relationship between HYSPLIT points and smoke quantity – is this purely based on duration assigned by the analyst or is land cover taken into account? Do all HYSPLIT points emit the same amount of smoke? This also raises the question again of why hours of smoke are being used instead of derived emissions.

The number of HYSPLIT Points analyzed, multiplied by their respective duration (Smoke Production Duration Hours or SPDH) is proportional to visual smoke produced for the land cover in which it is burning. This operational visual smoke analysis underpins our analysis, and this paper represents a different approach to CTM-based studies. However, this does not mean SPDH are proportional to emissions and we show this in Fig. 3c. Our primary objective is to estimate what fires are responsible for observed smoke plumes, attempting to derive emissions would add an additional layer of uncertainty to this analysis (uncertainties in emission factors, plant functional type, fuel moisture, combustion efficiency, weather conditions, fuel loading, *etc.*). SPDH reasonably distinguishes between fires that produce a small puff of smoke from fires that produce smoke plumes capable of covering entire states. We use HYSPLIT points because they are produced in tandem with the smoke plumes. You could not similarly use another emissions dataset to do this analysis. To answer the reviewer's question very directly, the relationship between smoke quantity (as defined by particulate matter concentration [$\mu g\ m^{-3}$] or

kg of emissions for example) and HYSPLIT points is variable and does not account for land cover.

The following sentence was added to section 2.1, the section that describes HYSPLIT points.

"The number of HYSPLIT points analyzed, multiplied by their respective duration (SPDH) is proportional to visual smoke produced for the land cover in which it is burning, and the analyst determines this relationship. The relationship between smoke quantity (as defined by particulate matter concentration [μg m$^{-3}$] or kg of emissions for example) and HYSPLIT points is variable and does not account for land cover."

The following sentences was added to the end of section 5.2, the section that defines smoke hours.

"A further weaknesses of smoke hours is that they have no way of accounting for secondary organic aerosol particle formation, which previous studies have shown can be significant (*e.g.* Sakamoto et al., 2015 & 2016; Janhall et al., 2010). However, these processes remain poorly understood and represented in chemical transport models, so an appreciation for the fact that these processes exist, but not attempting to account for them is sufficient for understanding the results presented in this work."

New citations (due to SOA discussion):
Sakamoto, K. M., J. D. Allan, H. Coe, J. W. Taylor, T. J. Duck, and J. R. Pierce. "Aged Boreal Biomass-Burning Aerosol Size Distributions  from BORTAS 2011." *Atmos. Chem. Phys.* 15, no. 4 (February 16, 2015): 1633–46. doi:10.5194/acp-15-1633-2015.
Sakamoto, K. M., J. R. Laing, R. G. Stevens, D. A. Jaffe, and J. R. Pierce. "The Evolution of Biomass-Burning Aerosol Size Distributions due to Coagulation: Dependence on Fire and Meteorological Details and Parameterization." *Atmos. Chem. Phys.* 16, no. 12 (June 24, 2016): 7709–24. doi:10.5194/acp-16-7709-2016.
Janhäll, S., M. O. Andreae, and U. Pöschl. "Biomass Burning Aerosol Emissions from Vegetation Fires: Particle Number and Mass Emission Factors and Size Distributions." *Atmos. Chem. Phys.* 10, no. 3 (February 9, 2010): 1427–39. doi:10.5194/acp-10-1427-2010.

7. P7, line 6: Is the difference in magnitude actually due to comparison between 'SPDH' hours and actual C emissions rather than "varying emission factors for different ecosystems"?

We agree that there are more factors that could contribute to this difference, though we expect that varying emission factors for different ecosystems can partially explain the observed differences. We have removed the reference to emission factors in this sentence, and we have added the following sentences instead:

"The GFED4s estimate of grams of carbon emitted is a function of the estimated burn area, estimated fire combustion efficiency, soil moisture, the estimated land cover type, fuel loading, and emission factors (Giglio et al., 2013; Randerson et al., 2012; van der Werf et al., 2017). The observed difference between SPDH and GFED4s total grams of carbon emitted could be due to any of these factors."

8. P9, line 30-31: Just a comment that a modelling approach would better allow altitude specific analysis to be conducted e.g. at surface level where air quality is of concern to the population.

To keep our analysis as useful as possible to others, we have chosen to avoid this. For most of the paper we did not set a limit on trajectory height, but we did link smoke plumes to elevated surface $PM_{2.5}$ in Fig. 1.

The shortcomings of chemical transport models ability to estimate ground level $PM_{2.5}$ resulting from smoke has been demonstrated repeatedly (*e.g.* Lassman et al., 2017; Reid et al., 2015; Alvarado et al., 2015; Paugam et al., 2016; Baker et al., 2016; Garcia-Menendez et al., 2013; Saide et al., 2015). Please see comments on a modelling approach listed above.

9. P11, line 2: It would be good to provide some explanation on why the trajectories run using the EDAS data are nearly identical and if this is something expected given the higher resolution of the meteorological input.

When we used the phrase 'nearly identical' we mean in terms of the impact of the interpretations of the results presented in the paper. Quantitatively the numbers are different, as shown in tables S1 and S2 and Fig. S4 - S6 in the Supplementary Information. The similarities of the results between the two different meteorology datasets show that our methods are not sensitive to the chosen meteorology, possibly since the results are dependent on long range transport, a spatial scale much greater than even the 1 degree GDAS data.

P11, line2: The sentence that used the phrase "nearly identical" has been changed to read:

"When the analysis presented in Fig. 11 is generated with trajectories run using EDAS meteorology data, the interpretation of the results is nearly identical and they are available in Sect S7 of the Supplemental Information."
* * *
All changes to the manuscript resulting from reviewer comments discussed above are shown in the pages that follow. We show the "track changes" so that it is easy to see what was changed due to reviewer comments/discussion.
* * *

[revised manuscript text omitted]

**S9.1 Comparison of 1990 AVHRR land cover data to 2010 MODIS/TERRA land cover data**

In this section we assign HYSPLIT points land cover using a dataset from the North American Land Cover Change Monitoring System. The specific dataset we choose to use within this system was the 2010 Land Cover of North America at 250 meters data published in 2013 by the Commission for Environmental Cooperation in Montral, Qubec, Canada (`http://www.cec.org/tools-and-resources/map-files/land-cover-2010`). These data are based on the Moderate Resolution Imaging Spectroradiometer (MODIS/TERRA) seven land spectral bands top of atmosphere reflectance. These data were created primarily to assess North America land cover changes between 2005 and 2010. Approximately 1% of land area land cover classifications changed during this time period.

Overall we find that the differences between the two datasets are driven by an inconsistent list of land cover classifications between the 1992 and 2010 data. Comparing the two different land cover classifications is challenging and requires subjectivity due to the fact that the land cover classification categories are not the same between the 1992 and 2010 data. For example, when assessing whether the two datasets agree on land cover for a given location there needs to be a decision as to whether "Temperate or subpolar needleleaf forest" and "Evergreen Needleleaf Forest" count as a match. Instead of making this decision we show what land cover assignment would have been made for each HYSPLIT point had we used the 2010 data rather than the 1992 data using the same methods described in Sect 2.3 of the manuscript. The only difference is that we allow "Urban" land cover assignments to be made, as these updated data were created after many of the fires in our analysis occurred.

Figures S8 through S24 show what the 2010 land cover data would have assigned to HYSPLIT points for each type of 1992 land cover classification made. For example, Fig. S8 shows the location of HYSPLIT points where the 1992 data assigned the land cover type "cropland/grassland mosaic" and color codes them by the land cover that would have been assigned had we used the 2010 data. The bar graphs on the left shows the abundance of each 2010 data classification.

[Figure]

Figure S8

[Figure]

Figure S9

[Figure]

Figure S10

[Figure]

Figure S11

[Figure]

Figure S12

[Figure]

Figure S13

[Figure]

Figure S14

[Figure]

Figure S15

[Figure]

Figure S16

[Figure]

Figure S17

[Figure]

Figure S18

[Figure]

Figure S19

[Figure]

Figure S20

[Figure]

Figure S21

[Figure]

Figure S22

[Figure]

Figure S23

[Figure]

Figure S24

**S9.2 Comparison of 1990 AVHRR land-cover data to Google Earth visible imagery**

The following weaknesses were observed when comparing the land cover assignment made using the methods described in Sect 2.3 of the manuscript to visible imagery provided by Google Earth. Steven Brey extensively audited the quality of the automated land cover assignments by plotting HYSPLIT point locations in Google Earth and using the visual imagery to make my own human assessment of the land cover. Determining plant species is not possible using this method, but it is possible to differentiate crops, grass, forest, water, and urban land cover types.

- Cropland and grassland seem to get mixed up in dry places. Heavily irrigated (green in visible Google Earth imagery) farmland appears to more regularly be classified as cropland. I observed that croplands in Eastern Colorado are often classified as grassland.

- The summit of Mount Rainier and Mount Baker are mixed forest. That is nonsense as these are heavily glaciated peaks and this dataset does have glaciers and snow cover. It correctly assigned snow and ice to the summit of a mountain in the Alaskan Range.

- There is also a consistent issue with assigning forest land cover classifications to agriculture in Western Washington. Because of the age of this dataset it is possible that this was indeed forest in the early 90s and has since been converted to cropland.

- At the interface between grass, shrubs and forests it takes a considerable distance to transition assignments to forest. For example, the data assign Arthurs Rock in Lory State Park Colorado, a forested area, as grassland.

- There are around ~10,000 HYSPLIT points (~1.3%) that are not given a land type assignment because the land cover data is convinced they are in urban areas or water, which my current methods do not allow. One example of when my methods fail is the HYSPLIT point that occurred on 2005-09-29 at (34.195, -118.259), near the middle of the Verdugo Mountains in Southern California. These mountains are less then 5 km across and are surrounded by expansive heavily developed cities that include Pasadena, Glendale, and Burbank California.

Overall the land cover assignments seem to make sensible assignments. For the purpose of distinguishing crops from forest from shrubs this dataset and methods described in Sect 2.3 of the manuscript appear affective.

**S10    Additional HMS smoke plume operational analysis details**

Presently, there is no comprehensive validation of HMS smoke plume analysis. One of the reasons in that there is no spatially and temporally comprehensive ground truth to compare to. Other satellite data would need to be used and these would have their own uncertainties. In the manuscript we state that the largest uncertainty for the smoke plume analysis is likely to be the edges of the smoke plume. In aggregate this is true, but may not be the largest uncertainty for every individual smoke plume. Below is an example of how the edge of a smoke plume can be uncertain, as described by an HMS analyst.

Description of smoke plume analysis on 9/23/2017: We [HMS analyst] observed agricultural burning in the mid/lower Mississippi Valley the day before (Fri 9/22/2017). When I was drawing my smoke on the morning of the 23rd I could see a remnant plume from the previous evening that was clearly associated with one of the larger fires and it was somewhere over southwest MO and vicinity. This plume was attached to a larger area of lighter density smoke that extended to the east into the Ohio Valley. As is typical in the eastern half of the US in the summer there was also sulfate haze. Since there was no hard edge on the eastern extent of the smoke I had to make a best guess as to how far the smoke extended. The problem is most pronounced with larger smoke plumes as they become more detached from the source fire since they gradually disperse and lose their sharp edge.

Many of the archived smoke polygons have straight-line edges particularly during the summer and over the ocean. The straight edges signify a boundary in which smoke-plume-detection analysis is performed. The smoke-plume-detection analysis is performed in five sectors. Each sector displays satellite imagery in a Lambert conic conformal projection. After analysis in all five regions has been performed, they are pieced together to form a single analysis. Strait edges of individual smoke GIS polygons occur when smoke plumes from different regions are pieced together. Not all sectors are analyzed year round. There is no analysis for Alaska or Northern Canada between November 1st and May 1st.

There is not always a HYSPLIT point associated with every smoke plume and vice versa. Often (especially during the summer when there are many wildfires producing a large amount of smoke) analysts observe smoke that has drifted a long way and become detached from the fire that produced it. In this case the smoke plume is not associated with any HYSPLIT points on that day. For example, the wildfires in Alaska and Northern Canada in 2015 produced smoke that drifted southeast into the Great Lakes and Mid Atlantic region and eventually reached Europe. This transport occurred over several weeks. When HMS analysts drew the smoke plumes as they traveled over the eastern part of the U.S., they did not associate the smoke with HYSPLIT points from the current day. There are also instances of HYSPLIT points when no smoke plumes are analyzed. For example, this can occur when HYSPLIT points are analyzed where there are many small fires, but no smoke plume analysis is done due to cloud cover.

**S11   Code repository:**

In an effort to be as open, transparent, and reproducible as possible, all work associated with this project is stored in the following subversion repository: `http://salix.atmos.colostate.edu/svn/smokeSource/`. This repository includes every version of all code, figures, and writing associated with this project. Please direct questions about this repository to sjbrey@rams.colostate.edu.

**S12   Interactive Version of Figure 13:**

A R Shiny application allows for interactive exploration of manuscript Fig. 13, including the ability to add or remove regions, as well as change between EDAS and GDAS meteorology. `https://stevenjoelbrey.shinyapps.io/smokeWheel/`

The Code for the app is located here: `https://github.com/stevenjoelbrey/smokeWheel`